# Genetically dissecting the electron transport chain of a soil bacterium reveals a generalizable mechanism for biological phenazine-1-carboxylic acid oxidation

**Lev M. Z. Tsypin**[1¤], **Scott H. Saunders**[2], **Allen W. Chen**[3], **Dianne K. Newman**[1,4]*

1 Division of Biology and Biological Engineering, California Institute of Technology, Pasadena, California, United States of America, 2 Green Center for Systems Biology—Lyda Hill Department of Bioinformatics, University of Texas Southwestern Medical Center, Dallas, Texas, United States of America, 3 Division of Chemistry and Chemical Engineering, California Institute of Technology, Pasadena, California, United States of America, 4 Division of Geological and Planetary Sciences, California Institute of Technology, Pasadena, California, United States of America

¤ Current address: Department of Pathology, Leland Stanford Jr. University, Palo Alto, California, United States of America

* dkn@caltech.edu

**Data Availability Statement:** The authors confirm that all data underlying the findings are fully available without restriction. All relevant data are

## Abstract

The capacity for bacterial extracellular electron transfer via secreted metabolites is widespread in natural, clinical, and industrial environments. Recently, we discovered the biological oxidation of phenazine-1-carboxylic acid (PCA), the first example of biological regeneration of a naturally produced extracellular electron shuttle. However, it remained unclear how PCA oxidation was catalyzed. Here, we report the mechanism, which we uncovered by genetically perturbing the branched electron transport chain (ETC) of the soil isolate *Citrobacter portucalensis* MBL. Biological PCA oxidation is coupled to anaerobic respiration with nitrate, fumarate, dimethyl sulfoxide, or trimethylamine-N-oxide as terminal electron acceptors. Genetically inactivating the catalytic subunits for all redundant complexes for a given terminal electron acceptor abolishes PCA oxidation. In the absence of quinones, PCA can still donate electrons to certain terminal reductases, albeit much less efficiently. In *C. portucalensis* MBL, PCA oxidation is largely driven by flux through the ETC, which suggests a generalizable mechanism that may be employed by any anaerobically respiring bacterium with an accessible cytoplasmic membrane. This model is supported by analogous genetic experiments during nitrate respiration by *Pseudomonas aeruginosa*.

## Author summary

Many bacteria have extremely flexible metabolisms, and we are only beginning to understand how they manifest in the environment. Our study focuses on the role of phenazine-1-carboxylic acid (PCA), a molecule that some bacteria synthesize and secrete into their surroundings. PCA is an "extracellular electron shuttle," a molecule that readily transfers

within the paper, its Supporting Information files and, as described in the Materials and Methods section, the entire dataset is available on the CaltechDATA repository, at the DOI https://doi.org/10.22002/tdng7-twd27. All data and code for analysis are available on GitHub (https://github.com/ltsypin/Cportucalensis_genetic_mech.

**Funding:** This work was supported by the NSF Graduate Research Fellowship to LMZT; by the NIH (1R01AI127850-01A1 to DKN) and by the Doren Family Foundation to DKN. The funders had no role in study design, data collection and analysis, decision to publish, or preparation of the manuscript.

**Competing interests:** The authors have declared that no competing interests exist.

**Abbreviations:** PCA, phenazine-1-carboxylic acid; ETC, electron transport chain; TEA, terminal electron acceptor; $NO_3^-$, nitrate; $NO_2^-$, nitrite; $N_2O$, nitrous oxide; NO, nitric oxide; $Fum^{2-}$, fumarate; $Succ^2$, succinate; DMSO, dimethyl sulfoxide; DMS, dimethyl sulfide; TMAO, trimethylamine-N-oxide; TMA, trimethylamine; UQ, ubiquinone; $UQH_2$, ubiquinol; MQ, menaquinone; $MQH_2$, menaquinol; DMQ, demethylmenaquinone; $DMQH_2$, demethylmenaquinol.

electrons between cells and oxidizing/reducing compounds or other cells. Until our investigation, the role of PCA electron-shuttling had only been studied in one direction: how it takes electrons away from cells, and the effect this has on their viability. Here we present a detailed account of the opposite process and its mechanism: what happens when PCA delivers electrons to cells? Our findings indicate that this previously underappreciated process is generalizable to any anaerobically respiring bacterium. Consequently, we expect that electron donation by PCA is widespread in environments where PCA is plentiful and oxygen is sparse, such as in some agricultural soils. The universality of the extracellular electron shuttle oxidation mechanism we describe for PCA suggests that it should also occur with similar small molecules, of which there are thousands, deepening the implication that this is a significant process in the environment and motivating further research into its consequences.

## Introduction

Phenazines are secreted secondary metabolites produced by diverse soil bacteria [1] that microbes use in various ways: from quorum sensing [2] to antimicrobial warfare [3–5] to energy conservation under anoxia [6]. Each of these biological roles is connected to the ability of phenazines to accept and donate electrons (i.e., their redox activity), a process that has been studied for over 120 years. Bacterial phenazine reduction was first proposed in the nineteenth century as an indicator for the presence of enterics in water supplies [7]. Several decades later, pyocyanin, one of the phenazines produced by *Pseudomonas aeruginosa*, was described as an "accessory respiratory pigment" that increased the rate of oxygen consumption by *Staphylococcus*, *Pneumococcus*, and erythrocytes by shuttling electrons from the cells to oxygen [8]. Once it became apparent that phenazines can have cytotoxic effects, they were characterized as antimicrobial compounds that destructively abstract electrons from the transport chain [3]. It was then discovered that reducing phenazines can greatly benefit *Pseudomonas aeruginosa* by 1) regulating gene expression during quorum sensing by oxidizing a transcription factor; 2) acting as alternative terminal electron acceptors to promote anoxic survival; and 3) facilitating nutrient acquisition [2,6,9–11] These reports paint a complex picture of the multifarious effects phenazines can have, but in each case, the conceptual model ends with the cell reducing the phenazine, which raises the question: how are phenazines recycled?

The first answer is: abiotically. Phenazines are broadly reactive molecules and can be oxidized by a variety of oxidants, including molecular oxygen and manganese or iron minerals. When oxygen serves as the electron acceptor, superoxide is produced, harming both phenazine producers and other cell types [12]. In contrast, when iron minerals, to which phosphate is adsorbed, serve as the electron acceptor, ferrous iron and phosphate can be released, alleviating nutrient limitation [11,13,14] However, not all oxidants of higher redox potential (e.g., nitrate and nitrite) react quickly enough with phenazines to re-oxidize them abiotically on biologically relevant timescales. Nonetheless, bacteria with versatile respiratory electron transport chains, such as the soil isolate *C. portucalensis* MBL, can catalyze biological phenazine oxidation during anaerobic respiration [15,16]. Moreover, the thermodynamics of biological phenazine oxidation with a variety of terminal electron acceptors are favorable (see our theoretical treatment of this subject in the supplement, S1 Text). These observations provide a second possible answer to the question of how phenazines are recycled: it stands to reason that in anoxic microenvironments where phenazine-reactive abiotic oxidants are limited, phenazine-reducing bacteria may benefit from the presence of phenazine-oxidizing bacteria.

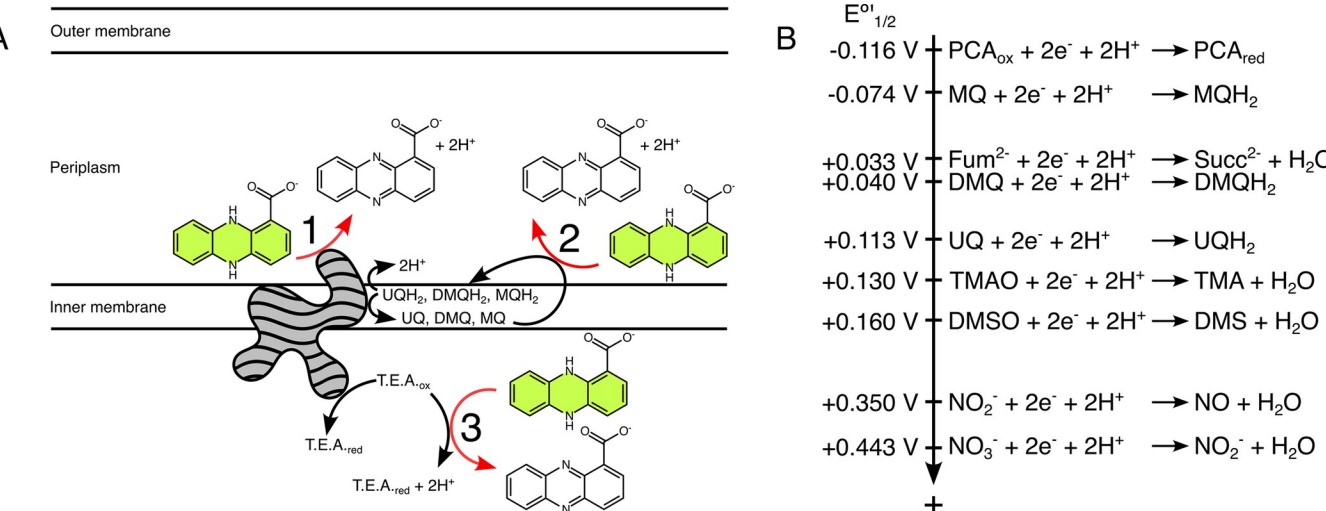

**Fig 1. Models and thermodynamics of PCA oxidation.** *(A) Potential models of PCA oxidation in Gram-negative bacteria capable of respiration.* In aqueous environments, PCA redox reactions are two-electron, two-proton processes. Reduced PCA is shown in green, reflecting its true color. PCA oxidation can theoretically be coupled to the respiratory electron transport chain in a couple ways: (Category 1, electron transfer to terminal reductase) PCA may donate electrons to the terminal reductase (grey shape) for a respirable terminal electron acceptor (T.E.A.), thus contributing two protons to the periplasm; (Category 2, electron transfer to quinol pool) PCA may donate two electrons and two protons to quinones, thus regenerating the quinol pool. (Category 3, electron transfer to terminal electron acceptor) Alternatively, PCA directly reduces the terminal electron acceptor. This may happen externally to the cell, or PCA may enter the cytoplasm and react with the terminal electron acceptor independently of the electron transport chain, as depicted. In this illustration, the arrows may represent direct reactions or ones mediated by enzymes or other factors. Transferring electrons to the terminal reductase or quinol pool (Categories 1 and 2) represents scenarios that may be energetically beneficial for a respiring bacterial cell, whereas transferring electrons to the terminal electron acceptor (Category 3) may be detrimental. This would require the transport of PCA across the inner membrane because its carboxylic acid moiety is negatively charged at circumneutral pH, and it cannot passively cross the membrane. For simplicity, this illustration does not show potential reactions with a periplasmic reductase, but the logic would remain the same, only with no involvement of the cytoplasmic space. *(B) Electron tower of relevant half-reactions.* Reactions are ordered by their relative standard midpoint potentials with more negative values on top and more positive ones on the bottom (not to scale). Thermodynamically favorable pairings comprise more positive half-reactions with more negative ones in reverse. The theoretical limit for energy that can be conserved from a pairing correlates with the magnitude of the difference in half-reaction potentials. PCA: phenazine-1-carboxylic acid. MQ: menaquinone. DMQ: demethylmenaquinone. UQ: ubiquinone. $Fum^{2-}$: fumarate. $Succ^{2-}$: succinate. TMAO: trimethylamine-N-oxide. TMA: trimethylamine. DMSO: dimethyl sulfoxide. DMS: dimethyl sulfide. $NO_2^-$: nitrite. NO: nitric oxide. $NO_3^-$: nitrate.

Furthermore, we hypothesize that phenazine oxidation itself might provide a survival advantage to phenazine-oxidizing cells under anaerobic conditions where electron donors are limited.

To test these ideas, we used a systematic genetic approach to dissect how *C. portucalensis* oxidizes phenazine-1-carboxylic acid (PCA), which is naturally produced, widespread, and environmentally relevant [17]. We focused on potential catalysts within the *C. portucalensis* electron transport chain (Figs 1A and 2, and Table 1). In addition to leading us to a generalizable mechanistic model for how biological PCA works, the development of a genetic system in this recently isolated soil organism exemplifies how the adaptation of existing tools can be used to rapidly gain new insights into microbial processes of environmental interest.

## Results

### Inactivation of individual nitrate reductases yields mild PCA oxidation defects

To investigate the mechanisms and dynamics of PCA oxidation by *C. portucalensis* MBL, we adapted *Escherichia coli* genetic engineering protocols [18,19]. Given that *C. portucalensis* harbors three functionally redundant nitrate reductase complexes (whose catalytic subunits

## A    Respiratory nitrate reductase

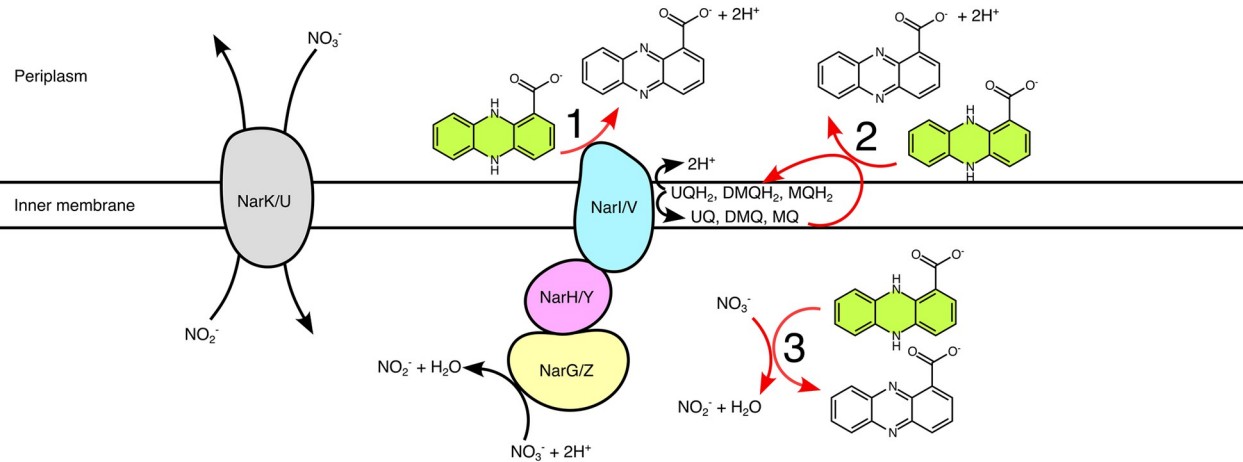

## B    Periplasmic nitrate reductase

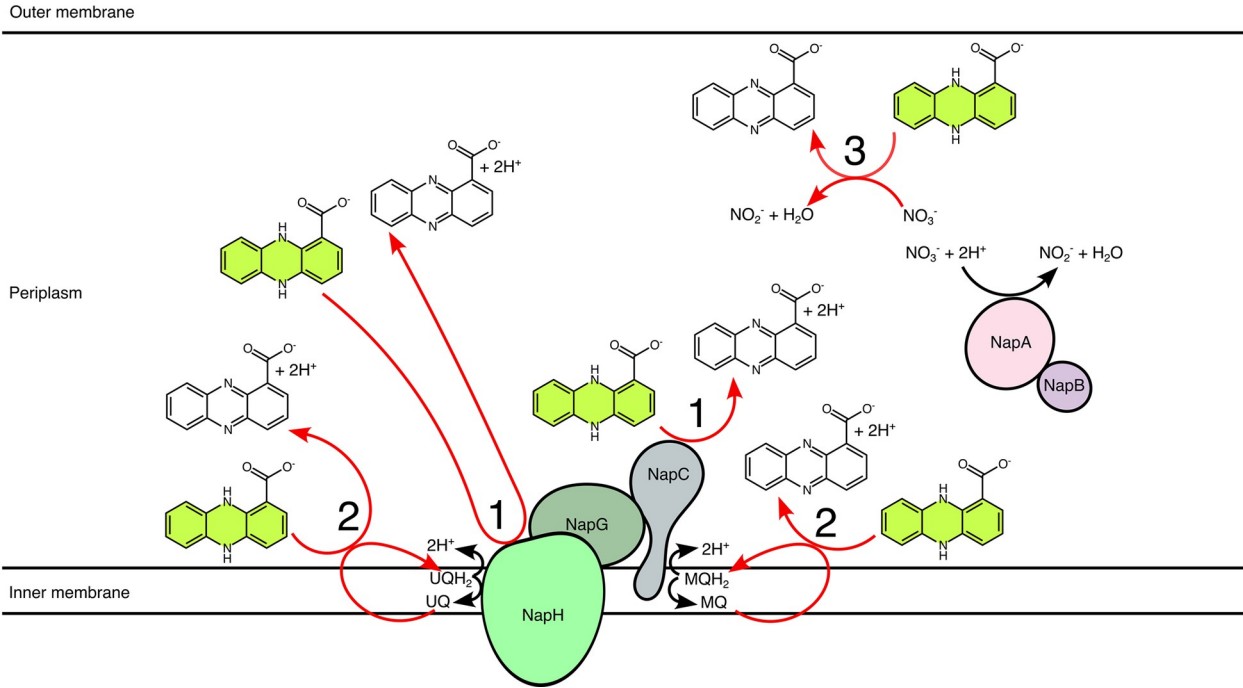

**Fig 2. Explicit conceptual model of PCA oxidation during anaerobic nitrate respiration.** To interpret PCA oxidation phenotypes during anaerobic nitrate respiration, it is necessary to keep in mind the distinct electron pathways in the respiratory (A) and periplasmic (B) nitrate reductase complexes. These models are made assuming a complete analogy to the arrangement of these proteins in *E. coli* [25, 29, 30]. The paths of the electrons along the reductase complexes are not shown for simplicity, but they flow from the quinols to the nitrate in each case. The categories of PCA interactions are numbered according to the scheme in Fig 1A and Table 1. In the case of the respiratory nitrate reductase complexes, there

are two redundant homologs in *C. portucalensis*: NarGHI and NarZYV [42]. NarI/V can accept electrons from all three types of quinones [25]. For the periplasmic nitrate reductase, there are two distinct quinone interaction sites (NapH for ubiquinone and NapC for menaquinone); notably, demethylmenaquinone does not appear to play a role in periplasmic nitrate reductase activity [30]. The NapAB complex is soluble in the periplasmic space, and the electrons from the NapHGC complex are ferried to NapA by NapB, which is a cytochrome c-type protein [29]. Category 1 (electron transfer to terminal reductase) PCA interactions are depicted as occurring at quinol-oxidizing subunits of the protein complexes (NarI, NarV, NapH, and NapC) to illustrate the hypothesis that a reduced PCA molecule may replace a quinol. Auxiliary and chaperone proteins that are members of the nitrate reductase operons and are involved in complex formation but not activity (NarJ, NarW, NapF, and NapD) are not illustrated [29]. Note: *P. aeruginosa* possesses only one set of homologs for the respiratory nitrate reductase (NarGHI) and the periplasmic nitrate reductase.

comprise NapA, NarG, or NarZ), we hypothesized that inactivating any one of these enzymes would be insufficient to abolish PCA-oxidation activity. To start, we compared the phenotypes of strains in which we perturbed nitrate reductases by two different methods: λRed recombination or oligo-mediated recombineering to make whole operon deletions or targeted translational knockouts of catalytic subunits, respectively [18,19]. Fig 2 shows the three nitrate reductases we mutagenized, as well as their predicted orientations in the ETC. According to this model, deleting the entire operon or knocking out just the catalytic subunit should have the same effect: all three categories of PCA oxidation reactions would be abolished.

Disrupting any single nitrate reductase by either genetic engineering method reduced but did not eliminate PCA oxidation. We observed no difference between the *narZ* translational knockout (*narZ-tlKO)* and the operon deletion (*ΔnarZYWV*) in the rate and dynamics of PCA oxidation (Fig 3A). Comparing *narZ-tlKO* and *ΔnarUZYWV*, we found that *ΔnarUZYWV* had a greater delay before PCA oxidation commenced, but the maximum rate of PCA oxidation was the same as for *narZ-tlKO* and *ΔnarZYWV* (Fig 3A). All three mutants oxidized PCA later and more slowly than the wildtype (Fig 3A). The lag in oxidation by the *ΔnarUZYWV* strain compared to the *ΔnarZYWV* and *narZ-tlKO* strains may reflect the fact that *narU* is an inner membrane nitrate-nitrite antiporter [20]: its absence may delay nitrate entrance to the cytosol where it can encounter NarZ or NarG (Fig 2). As observed for the *narZ* mutants, there was no difference between the *narG-tlKO* and *ΔnarGHJI* PCA oxidation phenotypes, and there was only mild loss of PCA reduction compared to the wildtype (Fig 3B). The *napA-tlKO* strain appeared to have a more severe PCA oxidation defect than *ΔnapFDAGHBC* versus the wildtype, yet loss of PCA oxidation by these strains still only minimally slowed down PCA oxidation (Fig 3C). For the rest of this study, we used oligo recombineering rather than λRed recombination to generate strains with multiple gene disruptions.

## PCA oxidation dynamics with nitrate depend on which nitrate reductases are present

Given the presence of three nitrate reductases in the genome, we speculated that our ability to alter PCA oxidation dynamics via their deletion, alone or in combination, would depend on their expression under different growth conditions. Accordingly, we compared nitrate-driven

**Table 1. The categories of PCA oxidation reactions.**

| Type | Description |
| --- | --- |
| Category 1 | PCA reduces the terminal reductase. |
| Category 2 | PCA reduces a quinone, thereby replenishing the quinol pool. |
| Category 3 | PCA reduces the terminal electron acceptor, either internally or externally to the cell (i.e., abiotic oxidation of PCA by the terminal electron acceptor). |

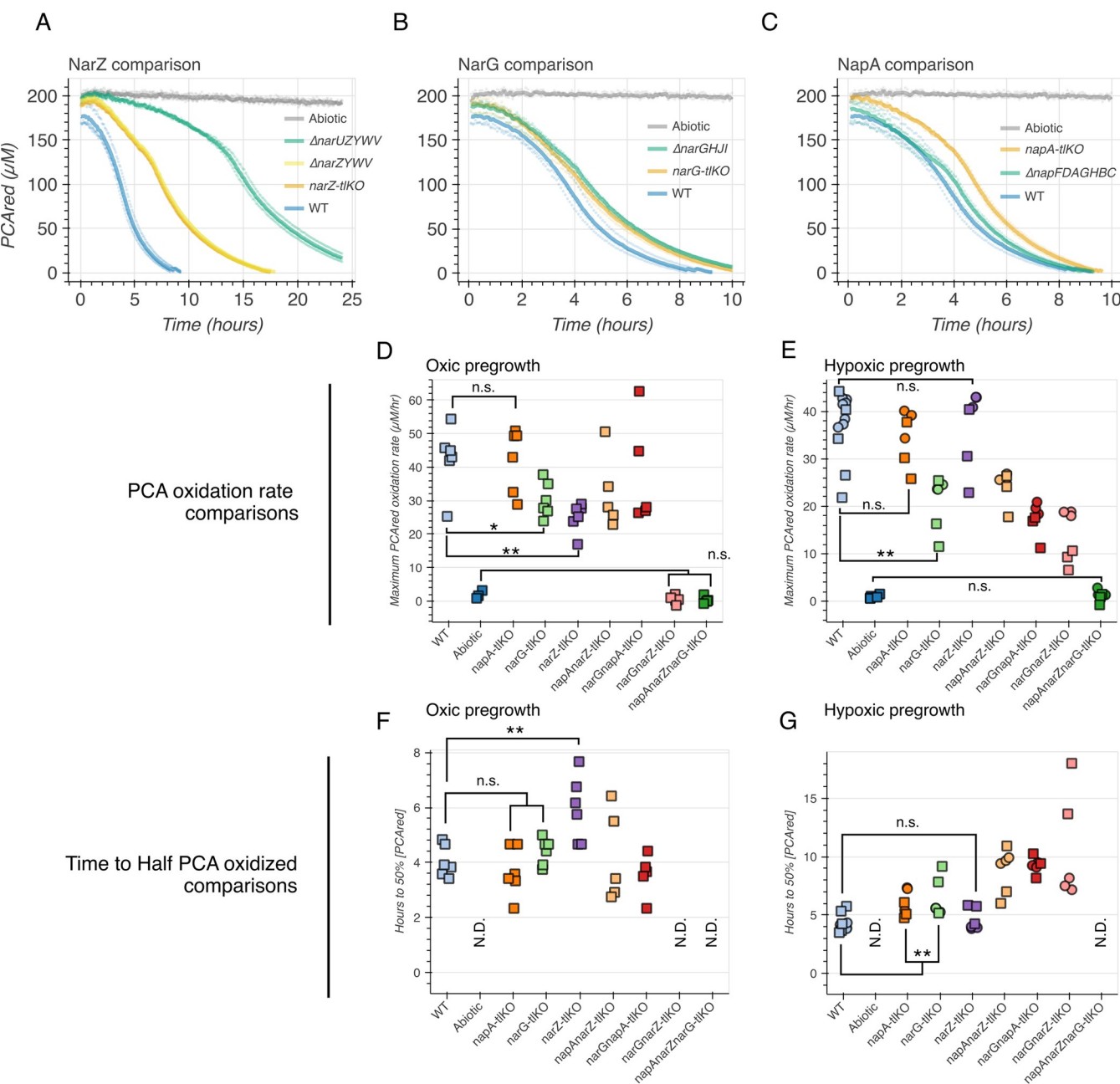

**Fig 3. Comparison of the roles of the three terminal nitrate reductases in PCA oxidation.** (A-C) Comparison of homologous recombination knockouts versus translational knockouts for each individual terminal nitrate reductase. Each graph shows the oxidation of PCA over time (measured by the decay of PCAred, which is fluorescent) with the abiotic control (nitrate plus reduced PCA in reaction medium without cells) in grey. (A) NarZ comparison: the deletion (Δ) and translational knockout (tlKO) strains all have a PCA oxidation deficit relative to the wildtype (WT, blue). The most severe phenotype is in the *ΔnarUZYWV* strain (green), and the *ΔnarZYWV* and *narZ*-tlKO (yellow and orange, respectively) phenotypes are indistinguishable. (B) NarG comparison: the deletion (green) and translational knockout (orange) strains have the same slight PCA oxidation deficit relative to the wildtype control (WT, blue). (C) NapA comparisons: the translational knockout (orange) has a more severe PCA oxidation deficit that the deletion strain (green), relative to the wildtype control (blue). Each thick line corresponds to the mean of three biological replicates plotted in semitransparent circles. (D-E) Comparisons of the maximum observed PCA oxidation rate for each combinatorial nitrate reductase mutant (and abiotic control). (D) The phenotypes after shaking (oxic) pregrowth. (E) The phenotypes after standing (hypoxic) pregrowth. (F-G) Comparisons of the time until half of the PCA was oxidized for each combinatorial nitrate reductase mutant (and abiotic control). (F) The phenotypes after oxic pregrowth. (G) The phenotypes after hypoxic pregrowth. N.D. stands for "not detected," corresponding to strains that did not reach the 50% PCA oxidized threshold over the 48-hour assay. Squares represent the means of technical triplicates and circles are independent biological replicates with no technical replicates. Double asterisks represent statistical significance after Bonferroni correction, and single asterisks represent p < 0.05 before Bonferroni correction. The statistical testing is described in the supplementary methods (S1 Text), and the complete matrices of pairwise comparisons are presented in S2 Fig. The conversion of PCA oxidation curves to rate and time metrics is described in S3 Fig.

PCA oxidation dynamics after two different pregrowth conditions in lysogeny broth (LB): slanted shaking overnight tubes and standing parafilm-sealed overnight tubes. These two conditions permitted fully aerated (oxic) and hypoxic growth, respectively. Depending on which cultivation condition was used, distinct nitrate reductases dominantly contributed to PCA oxidation (Fig 3D–3G).

After oxic pregrowth (Fig 3D and 3F), the *narZ* knockout had the most severe phenotype of the single mutants, and eliminating both *narG* and *narZ* was sufficient to abolish PCA oxidation, implying that under this condition *napA* is irrelevant (Fig 3D). The time it took for the different strains to oxidize 50% of the provided reduced PCA was correlated with the average maximum PCA oxidation rate, and the *narZ* single knockout was again the slowest among the other single mutants (Fig 3F). The abiotic, *narGnarZ* double knockout, and *napAnarZnarG* triple knockout conditions never reached this 50% threshold (Fig 3F). Relative to the wildtype, only the *narZ* single knockout strain had a significantly different time to 50% oxidation (Figs 3F, and S2A and S2C).

Cells pre-grown in stationary cultures provided a subtle but important contrast to the shaking pregrowth results (Fig 3E and 3G). Here, each of the double knockouts had a detectable PCA oxidation rate, indicating that any of the three nitrate reductases can drive the oxidation (Fig 3E). Moreover, rather than *narZ*, the greatest loss of oxidation in a single mutant background came from the *narG* (Fig 3E). Once again, the time to 50% oxidation was correlated to the maximum oxidation rate (Fig 3G), but in this condition, the *narZ* single mutant did not delay reaching this threshold relative to the wildtype activity (S2B and S2D Fig). These results imply that the pregrowth condition determines the relative presence and activity of the different nitrate reductases prior to the oxidation assay, leading to different phenotypes for a given strain. Because pre-growth in stationary cultures allowed us to observe the contributions of each of the three nitrate reductases, we employed this condition for the rest of the genetic experiments, including complementation assays (S4 Fig). When the nitrate reductases were individually overexpressed in the triple knockout background during overnight stationary pre-growth (S4A Fig), only the *narZ* overexpression strain had a statistically significant rescue of PCA oxidation (S4B Fig). The rescue effect was small, and the *napA* and *narG* overexpression strains did shift toward higher oxidation rates, so this is likely due to the overexpression system being unoptimized for *C. portucalensis* MBL. Summarily, PCA oxidation was abolished only when all three terminal nitrate reductases were knocked out.

The PCA oxidation dynamics and phenotypes can generally be summarized by the time it takes to reach a threshold concentration of reduced PCA (e.g., S3A Fig) and the maximal PCA oxidation rate (e.g., S3B Fig), but this can obscure some more nuanced phenotypes, such as the biphasic nature of many of the PCA oxidation curves (e.g., Fig 3A, *narZ-tlKO*). While the fit and oxidation metrics depend on the parametrization of the model, we were able to identify a conservative parametrization that robustly represented all tested strains (S3C–S3F Fig). For the remaining experiments, we report only the maximum PCA oxidation rates as the phenotypes of interest.

### Loss of quinones in C. portucalensis *MBL significantly disrupts PCA oxidation, yet mild oxidation is still achieved by nitrate reductases in their absence*

Nitrate-driven PCA oxidation by *C. portucalensis* is fully abolished only when all three terminal nitrate reductases are knocked out; any one remaining reductase enables PCA oxidation (Fig 4D). This observation raises the question whether the nitrate reductases oxidize PCA directly (Fig 2 and Table 1, Category 1 reactions) or if PCA contributes electrons to an

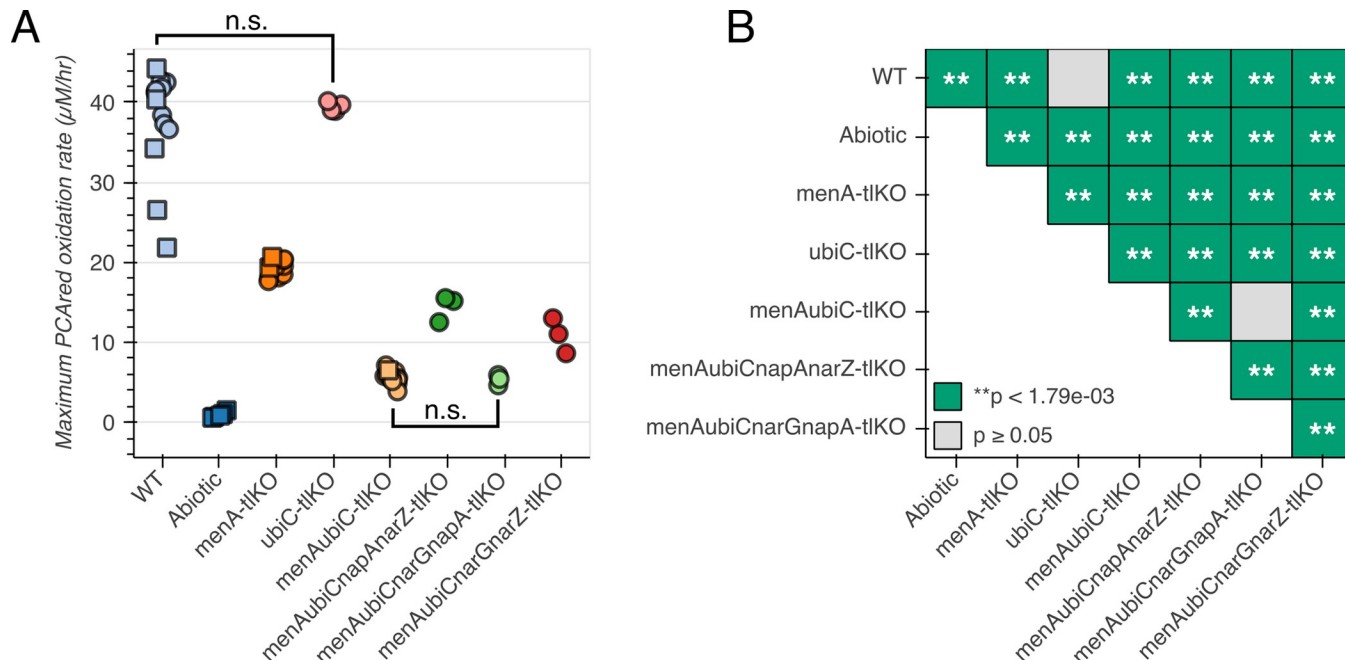

**Fig 4. Nitrate-driven PCA oxidation rates in quinone knockout backgrounds, including single nitrate reductase strains.** (A) Maximum PCA oxidation rates. Squares represent the means of technical triplicates and circles represent independent biological replicates. (B) Pairwise comparisons between differences of mean maximum PCA oxidation rates for the strains in (A). Given 28 comparisons, the Bonferroni-corrected threshold for significance is p < 0.00179.

upstream pool, such as the quinols (Fig 2 and Table 1, Category 2 reactions). PCA oxidation was largely lost when the biosynthesis of all three quinones (ubiquinone, menaquinone, and demethylmenaquinone) was disrupted in the *menAubiC*-tlKO strain, indicating that PCA oxidation by quinones is the main component of the PCA oxidation rate (Fig 4A). However, the PCA oxidation rate in this genetic background did not drop to abiotic levels, indicating that cellular nitrate reduction can drive PCA oxidation at a low rate without quinones as intermediaries (Fig 4A). In other words, PCA oxidation by terminal reductases also can occur. The loss of ubiquinones alone in the *ubiC-tlKO* background did not affect PCA oxidation, but the loss of (demethyl)menaquinones in the *menA-tlKO* strain did have an effect (Fig 4A and 4B). In the quinone-null background, PCA oxidation persists with any one of the nitrate reductases (Fig 4A), as evidenced by the positive oxidation rate in the *menAubiCnapAnarZ-*, *menAubiCnarGnapA-*, and *menAubiCnarGnarZ-tlKO* strains. Interestingly, the strains with no quinones and only intact *NarG* or *NapA* had faster PCA oxidation rates than the *menAubiC* strain on its own (Fig 4A and 4B). The co-occurrence of PCA transferring electrons to both terminal reductases and quinones (Category 1 and 2 reactions) implies that PCA oxidation may be plastic: it does not necessarily depend on a specifically evolved enzyme or pathway to proceed.

We were unable to genetically complement the loss of quinones during nitrate-driven PCA oxidation (S5A and S5B Fig). Curiously, overexpressing *menA* exacerbated the *menAubiC-tlKO* phenotype (S5A Fig). Because each nitrate reductase retains some PCA oxidation activity even in the absence of quinones (Fig 4A), it may be the case that exogenously expressing a low level of quinones is not enough to give a signal over the independent nitrate reductase activity. However, our greater degree of success with complementing *menA* for the other terminal

electron acceptors indicates that the system works in principle (see the alternative TEA section, below).

## Pseudomonas aeruginosa *PA14 replicates nitrate reductase knockout phenotypes from* C. portucalensis *MBL*

Given the apparent generalizability of PCA oxidation to the three nitrate reductases in *C. portucalensis* MBL and the observation that other bacteria perform the same metabolism (15,29), we assessed whether *P. aeruginosa* PA14 nitrate reductase mutants conform to the same mechanistic model. *P. aeruginosa* is particularly relevant as a point of comparison because it biosynthesizes PCA, has been studied extensively as a PCA reducer, and, like *C. portucalensis* MBL, also has both respiratory and periplasmic nitrate reductases (albeit only one homolog of the respiratory nitrate reductase complex, NarGHI) (Fig 2) [6,21,22]. We compared wildtype, *ΔnarG*, and *ΔnapAB* strains of *P. aeruginosa* PA14 after pregrowth in shaking and standing conditions, as we had done for *C. portucalensis* MBL. We observed that in the shaking pregrowth condition, the *ΔnarG* strain had no phenotype versus the wildtype, while the *ΔnapAB* strain had a severe, though incomplete, PCA oxidation defect (Fig 5, left). In contrast, after either of the standing pregrowth treatments, the *ΔnapAB* strain did not substantially differ from the wildtype, while the *ΔnarG* strain had severe, though incomplete, loss of PCA oxidation (Fig 5, right). This pattern of distinct nitrate reductases dominating PCA oxidation activity depending on the pre-growth condition corresponds to the fact that *napA* and *narG* are regulated by distinct systems: RpoS and Anr, respectively [23,24], meaning that the *P. aeruginosa napA* gene is regulated like the *C. portucalensis narZ* gene. The *P. aeruginosa* phenotypes thus correspond to the difference we observed between *C. portucalensis* MBL *narG* and *narZ* translational knockouts depending on pre-growth conditions (Fig 4C and 4D), which implies that the PCA oxidation mechanism is not specific to a given organism or enzyme but rather the architecture of the electron transport chain.

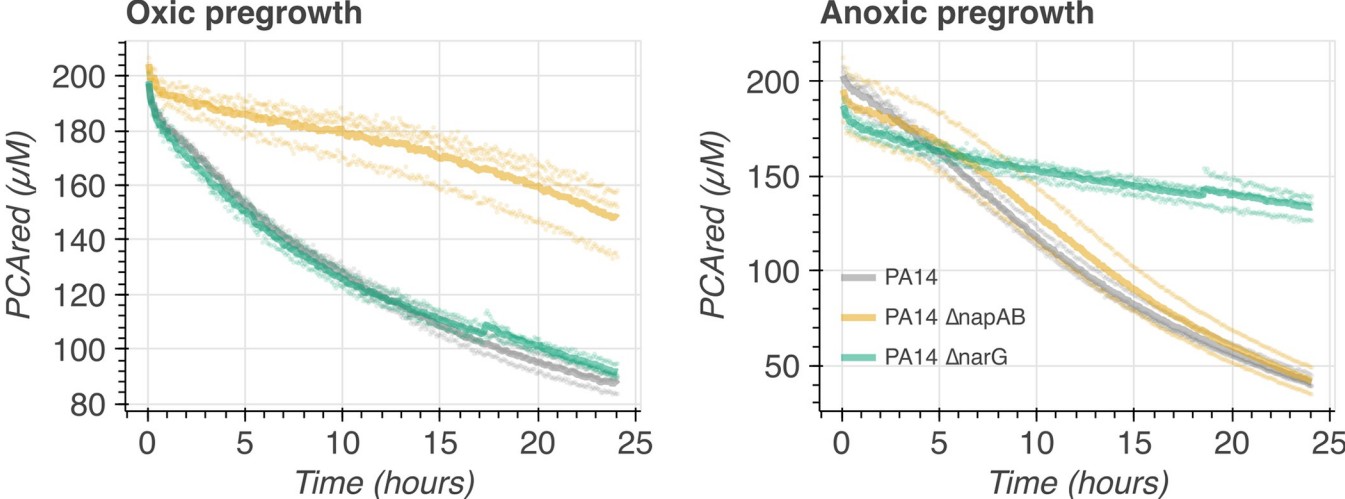

**Fig 5. Pre-growth dependent phenotypes for P. aeruginosa nitrate reductase mutants.** Left: When cultures were pre-grown in slanted shaking tubes, providing the cultures ample oxygen, only the *ΔnapAB* strain (orange) had a PCA oxidation deficit. Right: Pre-growing the cultures in standing tubes without nitrate supplementation also leads to the *ΔnarG* strain (green) to have the PCA oxidation deficit; the wildtype (grey) and *ΔnapAB* (orange) strains are within error of each other. Each thick line corresponds to the mean of three biological replicates that are plotted in semitransparent circles. The medium for this experiment was 84% LB and 16% basal PCA oxidation assay medium (see materials and methods). Note: due to its relative inability to grow under hypoxia in LB, it was not feasible to grow a *nap/nar* double knockout P. aeruginosa strain to compare to the *C. portucalensis* MBL full nitrate reductase knockout. Similarly, we could not generate any quinone knockouts for *P. aeruginosa* because it encodes the biosynthesis of only ubiquinone (24).

## Alternative TEAs drive PCA oxidation analogously to nitrate

Having found a generalizable mechanism for nitrate-driven PCA oxidation in *C. portucalensis* MBL, we tested whether it extended to the other anaerobic respiratory metabolisms. We found that fumarate (terminal reductase catalytic subunit FrdA), DMSO (terminal reductase catalytic subunit DmsA), and TMAO (terminal reductase catalytic subunit TorA) can all drive PCA oxidation by *C. portucalensis* MBL (Fig 6). Analogously to nitrate-driven PCA oxidation, there was no abiotic reaction between reduced PCA and these alternative terminal electron acceptors. Knocking out the terminal reductase was sufficient to abolish fumarate-driven PCA oxidation (Fig 6A). In fact, the *frdA* knockout strain quenched residual oxygen, thus lowering the apparent oxidation rate beyond that of the abiotic control (Figs 6A and S6A). The *menA* knockout strain, lacking (demethyl-)menaquinones lost some PCA oxidation activity, but the *ubiC* knockout strain was indistinguishable from the wildtype (Fig 6A). Unlike in the case of nitrate-driven PCA oxidation (Fig 4A and 4B), the *menAubiC* knockout strain did not have a more severe phenotype than the *menA* knockout alone, which corresponds to the standard model that the fumarate reductase does not engage ubiquinone [25]. It is also consistent that MenA complementation strain rescues some activity, and the UbiC complementation strain does not. In the complementation assays, the *frdA* overexpression vector was able to rescue roughly half of the wildtype activity over the *frdA* knockout background (Fig 6A, *frdA-tlKO* vs. *frdA* pFE21-FrdA), and the *menA* overexpression vector rescued PCA oxidation rates entirely (Figs 6A and S6A, WT vs. *menAubiC-tlKO* vs. *menAubiC* pFE21-MenA). The *ubiC* overexpression vector gave no effect (Figs 6A and S6A, *menAubiC-tlKO* vs. *menAubiC* pFE21-UbiC). Overall, the results here are analogous to what we observed while exploring nitrate-driven PCA oxidation: the abiotic control rules out Category 3 (electron transfer to terminal electron acceptor) PCA oxidation, the *frdA* knockout abolishes PCA oxidation entirely (just as the *napAnarZnarG* triple knockout did for nitrate), and the full loss of quinones retains partial PCA oxidation activity, which indicates that both Category 1 and 2 PCA oxidation reactions play a role.

DMSO-driven PCA oxidation followed largely the same patterns as for nitrate and fumarate, with the surprising exception of ubiquinone phenotypes. There was no abiotic activity

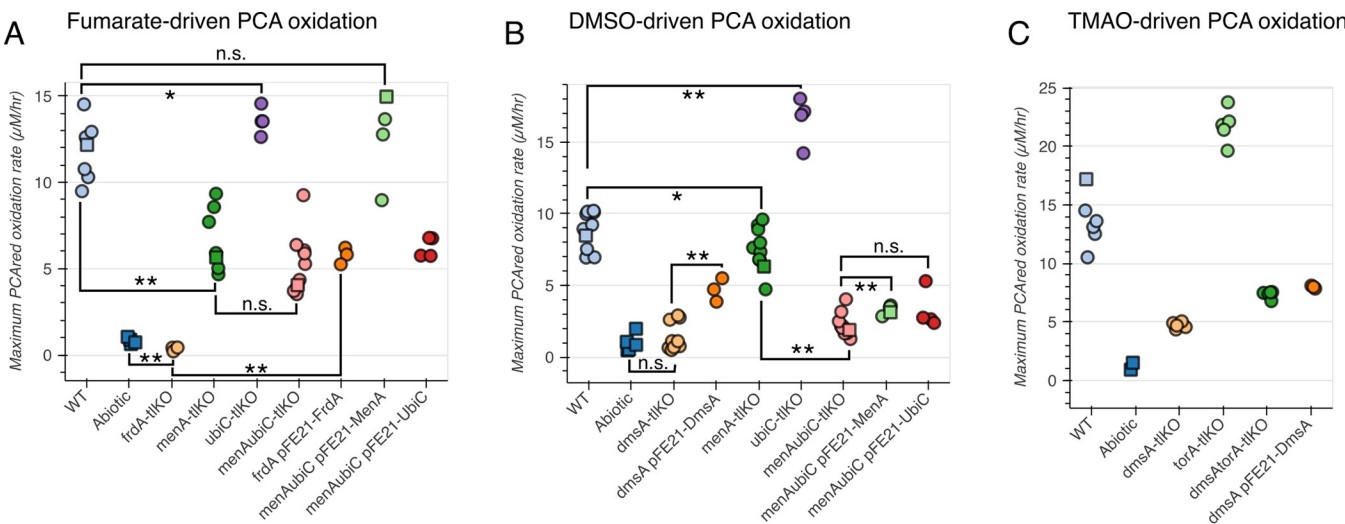

**Fig 6. PCA oxidation driven by alterative TEAs in their respective reductase and quinone knockout strains, as compared to corresponding complementations.** (A) Maximum PCA oxidation rate in the presence of fumarate. (B) Maximum PCA oxidation rates in the presence of DMSO. (C) Maximum PCA oxidation rates in the presence of TMAO.

with DMSO, ruling out Category 3 PCA oxidation (Fig 6B). The *dmsA* knockout lost all activity, which was partially complemented by the DmsA overexpression strain (Figs 6B and S6B). The *menA* knockout did not have a strong phenotype, but the *ubiC* knockout had a substantially faster PCA oxidation than the wildtype (Figs 6B and S6B). However, the *menAubiC* double knockout had a severe loss of PCA oxidation, though still measurable above the abiotic control (Figs 6B and S6B). The MenA complementation strain slightly rescued PCA oxidation relative to the *menAubiC* background, but the UbiC complementation strain had no rescue. These results again support the roles of both Category 1 and 2 PCA oxidation reactions, but our model does not explain why the loss of ubiquinones alone would increase PCA oxidation, while loss of ubiquinones on top of (demethyl-)menaquinones would decrease it.

Our exploration of TMAO-driven PCA oxidation is less complete because we were not able to fully abolish PCA oxidation by knocking out the catalytic subunits of terminal reductases. Knocking out *torA* increased the rate of PCA oxidation relative to the wildtype (Figs 6C and S6C). We found that the *dmsA* knockout strain had a PCA oxidation deficit during TMAO-driven PCA oxidation, which is consistent with the Dms complex promiscuously reducing many N-oxides [26,27] (Fig 6C). The *dmsAtorA* double knockout strain was faster at oxidizing PCA than the *dmsA* single knockout strain, implying that *C. portucalensis* MBL possesses other TMAO reductases. We did not pursue a full study of the role of quinones during TMAO-driven PCA oxidation because we could not constrain the full activity.

### Nitrate-driven PCA oxidation provides a survival benefit for C. portucalensis *MBL in a bioelectrochemical reactor*

With a new understanding of the mechanism of biological PCA oxidation, we can assess its impact on bacterial fitness over longer time scales. After having grown to stationary phase in LB, *C. portucalensis* MBL continuously oxidizes PCA, so long as nitrate is available in a bioelectrochemical chamber with a working electrode poised to a reducing voltage (Fig 7). Upon spiking fresh nitrate into the reactor after PCA oxidation ceases, the oxidation immediately resumes, indicating that nitrate availability is the limiting factor for these cultures (S7 Fig). Adding more nitrate while PCA oxidation was active does not affect the current, which implies that the minimum threshold for the permissible nitrate concentration is relatively low (S7 Fig, green trace). After several days of oxidizing PCA with nitrate, the apparent rate of nitrate consumption goes down, as evidenced by a persistent current in the 24-hour spiking condition after the third day (S7 Fig, yellow trace). In the cultures that received additional nitrate, there was a slight but steady decrease in total current over time (S7 Fig, yellow and green traces). We did not assess whether there are analogous dynamics for PCA oxidation in the bioelectrochemical chamber when provided any of the other terminal electron acceptors that we compared above.

Having observed PCA oxidizing activity over multiple days, we compared three potential metrics of *C. portucalensis* MBL fitness (colony forming units (CFUs), cellular ATP content, and nitrite production) in two chamber bioelectrochemical reactor conditions, with and without PCA, both providing a reducing working electrode and an initial concentration of 10 mM nitrate as the terminal electron acceptor (Fig 7B–7E). In the chamber with PCA, the cells converted nitrate to nitrite faster than in the chamber without PCA, which coincided with the loss of current as nitrite concentrations reached their asymptote (Fig 7B and 7D, third and fourth time points). Despite the PCA oxidation current ceasing between days one and three, the effect on survival was not apparent until day four, on which the cultures with PCA began to die (Fig 7C). Throughout the time course, there were no samples that indicated a difference in cellular ATP content between the two conditions (Fig 7E). Although this experiment cannot speak to

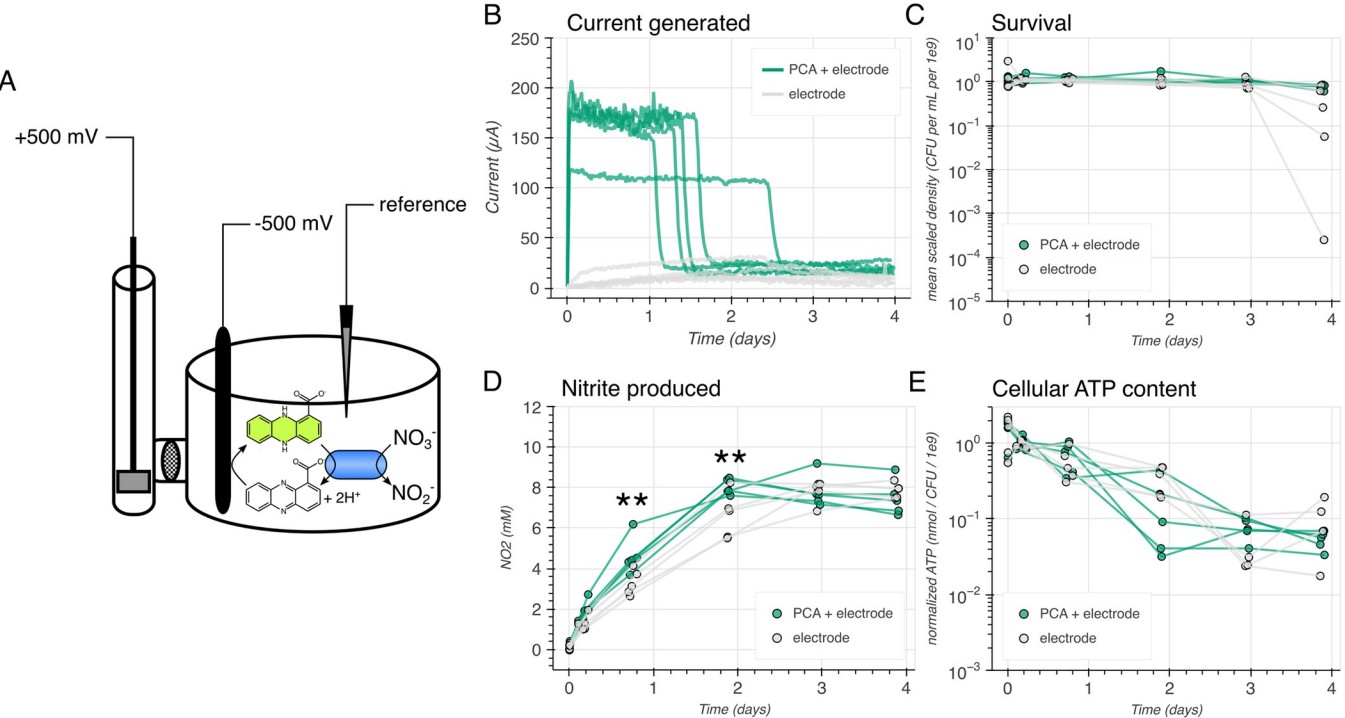

**Fig 7. Physiological consequences of long-term PCA oxidation for C. portucalensis MBL.** (A) Diagram of the bioelectrochemical reactor. The *C. portucalensis* MBL culture was incubated in the main chamber with a working electrode poised to -500 mV that continuously reduced any available PCA. The reference electrode communicated with the potentiostat to retain a constant voltage. Cells oxidized PCA when nitrate was present. In a sidearm, separated by a dense glass frit, the counter-electrode completed the circuit. Reduced PCA is green and oxidized PCA is colorless. (B) Current traces for biological replicates with and without provided PCA and an initial concentration of 10 mM nitrate. Samples were taken from these replicates at the times displayed in (C)-(E), where each data point corresponds to an independent biological replicate. (C) Time point measurements of survival in the culture, as determined by colony forming units (CFUs). The asterisks represent a Bonferroni-corrected statistically significant difference between the two conditions at the final time point (p = 0.0025). (D) The nitrite that was produced (from nitrate reduction) by the cultures over the experiment. At the third and fourth time point there were Bonferroni-corrected statistically significant differences between the two conditions: p = 0.0029 and p = 0.0019, respectively. (E) The normalized ATP content per CFU over the time course. There were no statistically significant differences. The Bonferroni corrected threshold was p < 0.0083 given hypothesis testing across the six time points.

the underlying reason for differential cell survival, it is apparent that extended PCA oxidation was beneficial under these conditions, in which PCA was the sole electron donor provided.

## Discussion

Our observations that (i) any respirable terminal electron acceptor stimulates PCA oxidation and (ii) that the presence of quinones is required for a large fraction of the oxidation rate imply a generalizable mechanism: any cell that can perform anaerobic respiration and has electron flux through the quinol pool should oxidize PCA. This is reflected in our inability to find a bacterium that does not oxidize PCA under suitable conditions [15,28]. Notably, had we not compared PCA oxidation by *C. portucalensis* MBL under differing pre-growth conditions, we may have had arrived at an errant model—linking PCA oxidation to particular terminal reductases. For example, had we relied on oxic-pregrowth cultures, we would have concluded that the periplasmic nitrate reductase is not involved in PCA oxidation (Fig 4C), yet it is now clear that multiple reductases can promote PCA oxidation by driving flux through the ETC when conditions are right for them to be active.

When comparing how different organisms and different terminal electron acceptors induce PCA oxidation, it is important to keep in mind the biosynthetic capacity of the organism (both

the presence of the pathway and its regulation) to make the various quinones (S1 Fig), as well as the ability of a given terminal reductase to use each of the quinones. For instance, *C. portucalensis* MBL synthesizes ubiquinone, menaquinone, and demethylmenaquinone, but the production of those quinones is differentially regulated depending on aerobic or anaerobic growth, and the periplasmic nitrate reductase only makes use of ubiquinone and menaquinone (not demethylmenaquinone) (Figs 2 and S1) [25,29,30]. Meanwhile, *Pseudomonas aeruginosa* PA14 can produce *only* ubiquinone, has a very limited capacity for fermentative growth and can only use nitrate for anaerobic respiration (S1 Fig). These features of *P. aeruginosa* PA14 made it unfeasible to test quinone-null or double nitrate reductase mutants [31,32], but they illuminated the generalizable principles of anaerobic PCA oxidation.

Species-specific quirks accounted for, the theoretical and experimental framework we describe in this report leads to several testable predictions (see the supplement, S1 Text). First, it forecasts the genetic and chemical conditions under which other phenazines such as pyocyanin (PYO, $E^{\circ'}_{1/2}$ = -0.040 V) would be oxidized [14]. PYO's redox midpoint potential, being more positive than that of PCA, indicates that its oxidation could be mediated by ubiquinone or demethylmenaquinone but not menaquinone (Fig 2B). For the same reason, fumarate should not be a suitable terminal electron acceptor to drive PYO oxidation. A knockout of the gene *ubiE*, which disrupts ubiquinone and menaquinone synthesis but keeps demethylmenaquinone synthesis intact (S1 Fig), would enable a more detailed exploration of the interactions of phenazines, quinones, and the anaerobically respiring electron transport chain [33,34]. While the *ubiCmenA* double knockout had a more severe PCA oxidation phenotype than the *menA* knockout alone, the *ubiC* single knockout phenotypes and the complementation results suggest that (demethyl-)menaquinones play a more significant role than ubiquinones (e.g., Fig 4A). Recent work from our lab demonstrated that ubiquinone spontaneously and instantaneously oxidizes PCA in aqueous solution [22]; the results presented here imply that the plasma membrane environment changes the dynamics of how phenazines and quinones interact. This motivates further dissection of the role of specific quinones during *in vivo* PCA oxidation in future studies.

We observed that, in the absence of quinones, PCA oxidation can still be driven by terminal reductases in the case of nitrate, fumarate, and DMSO. In other words, PCA electron transfer to terminal reductases may play a subtle, but significant role during anaerobic respiration. This may mean that either PCA is functionally substituting for a quinone, that there is an unknown other intermediate between PCA and the terminal reductase, or that there is an unappreciated abiotic oxidation of PCA in these contexts. The last case is unlikely: while nitrite that is produced during nitrate respiration can oxidize PCA at a low rate [15], its production requires that electrons be donated to the terminal nitrate reductase; without quinones, the next logical source of electrons would be the PCA, meaning that the PCA is ultimately oxidized by the terminal reductase and not nitrite. Furthermore, the fumarate and DMSO observations do not provide the same abiotic recourse as nitrite. There may well be an unknown factor that mediates electron transfer between PCA and the terminal nitrate and fumarate reductases. However, the existence of archaeal methanophenazine suggests that such a factor is not necessary. Methanophenazine is effectively the core phenazine molecule with a long aliphatic tail and is the membrane electron carrier in *Methanosarcina mazei*, which lacks quinones [35]. In other words, *M. mazei* provides an example of canonical electron shuttles being completely replaced by a phenazine in the membrane. Given this example and the fact that five different terminal reductase complexes catalyze PCA oxidation in the absence of quinones, the most parsimonious explanation is that reduced PCA replaces quinols as an electron donor to these complexes at an appreciable rate. The efficacy with which different phenazines may substitute for quinones likely depends on their hydrophobicity and diffusion within the plasma membrane.

The flexibility of conditions under which *C. portucalensis* MBL oxidizes PCA is under-scored by the functional redundancy of the various quinones and of certain terminal reduc-tases. This is most evident in the terminal nitrate reductases, where NarG, NarZ, and NapA each contribute different fractions of PCA oxidation depending on how the cultures were pre-grown. Assuming that the regulation of these genes is analogous to their orthologs' regulation in *E. coli*, NarG and NapA are regulated primarily by Fnr in response to hypoxia and nitrate availability. In *E. coli*, NarZ is primarily regulated by RpoS in response to growth arrest. This regulatory schema corresponds to our observation that the *narZ* single knockout has the most severe phenotype in the slanted shaking pre-growth treatment (in which cultures reach sta-tionary phase) and that the *narG* single knockout has the most severe phenotypes in the stand-ing pre-growth treatment (Fig 4). This is further supported by our comparison of the *nap* and *nar* knockouts in *Pseudomonas aeruginosa*. Thus, the ability of a cell to oxidize PCA is defined by both its genetic content and regulatory state. The redundant pathways for PCA oxidation enable organisms like *C. portucalensis* MBL to perform the process under many different envi-ronmental conditions, all of which have in common electron flux through the electron trans-port chain and the quinone pool.

In the case of TMAO-driven PCA oxidation, we were unable to fully abolish the activity by knocking out terminal reductases. We found that the DMSO reductase, as represented by the catalytic subunit DmsA, contributed to PCA oxidation by TMAO (Fig 7G). This corresponds to the promiscuity of the DMSO reductase that has been reported in other organisms [27]. However, the *torA* knockout had no phenotype on its own, and the *dmsAtorA* double knock-out oxidized PCA faster than the *dmsA* knockout alone (Fig 7G). This implies that there is another TMAO reductase(s) in *C. portucalensis* MBL that also participates in TMAO-driven PCA oxidation. Indeed, *C. portucalensis* MBL has two homologs to the catalytic subunit of the *E. coli* TorYZ reductase (NCBI accession IDs NUH53377.1 and NUH54782.1), which may be the culprits [36]. If knocking out these two genes in addition to *dmsA* and *torA* is not sufficient to abolish PCA oxidation activity with TMAO, the causative enzyme could potentially be iden-tified via a mutant screen in the quadruple knockout genetic background, looking for loss of PCA oxidation activity during TMAO respiration.

In addition to illuminating the molecular factors responsible for PCA oxidation, our work suggests that PCA oxidation may provide a fitness benefit for *C. portucalensis* MBL under some conditions (Fig 7). We presume that such a benefit would most likely arise when cells are starved for electron donors; in future follow-up studies, depletion of cellular internal prior to a survival assay would be expected to magnify a survival benefit. From our results, it is not clear how PCA oxidation promotes survival, as there was no effect on cellular ATP content over our experimental time period (Fig 7E), yet given its interactions with ETC components, it seems plausible that it could beneficially modulate membrane properties. Whether this is in fact the case, and whether PCA oxidation can support the growth of autotrophic organisms that rely on exogenous electron donors to fix inorganic carbon, are interesting physiological puzzles for future research.

Regardless, given that PCA reduction can provide a fitness benefit for *Pseudomonas aerugi-nosa* [6,37], it may be possible to pair phenazine oxidizing and reducing bacteria, such that the complete PCA redox cycle supports the survival of one or both (Fig 8A) [38]. Beyond employ-ing PCA oxidation to conserve energy, it is also possible for a bacterium to use PCA oxidation to compete against its neighbors (Fig 8B and 8C). Given that some bacteria appear to reduce PCA to release bioavailable iron and phosphorus (Fig 8B) [11,14], if an organism like *C. portu-calensis* MBL were to intercept and oxidize the PCA before it reached the target mineral, the PCA reducer would remain starved for iron or phosphorus (Fig 8C). Thus, depending on the context, PCA oxidation may serve either mutualistic or competitive interactions between

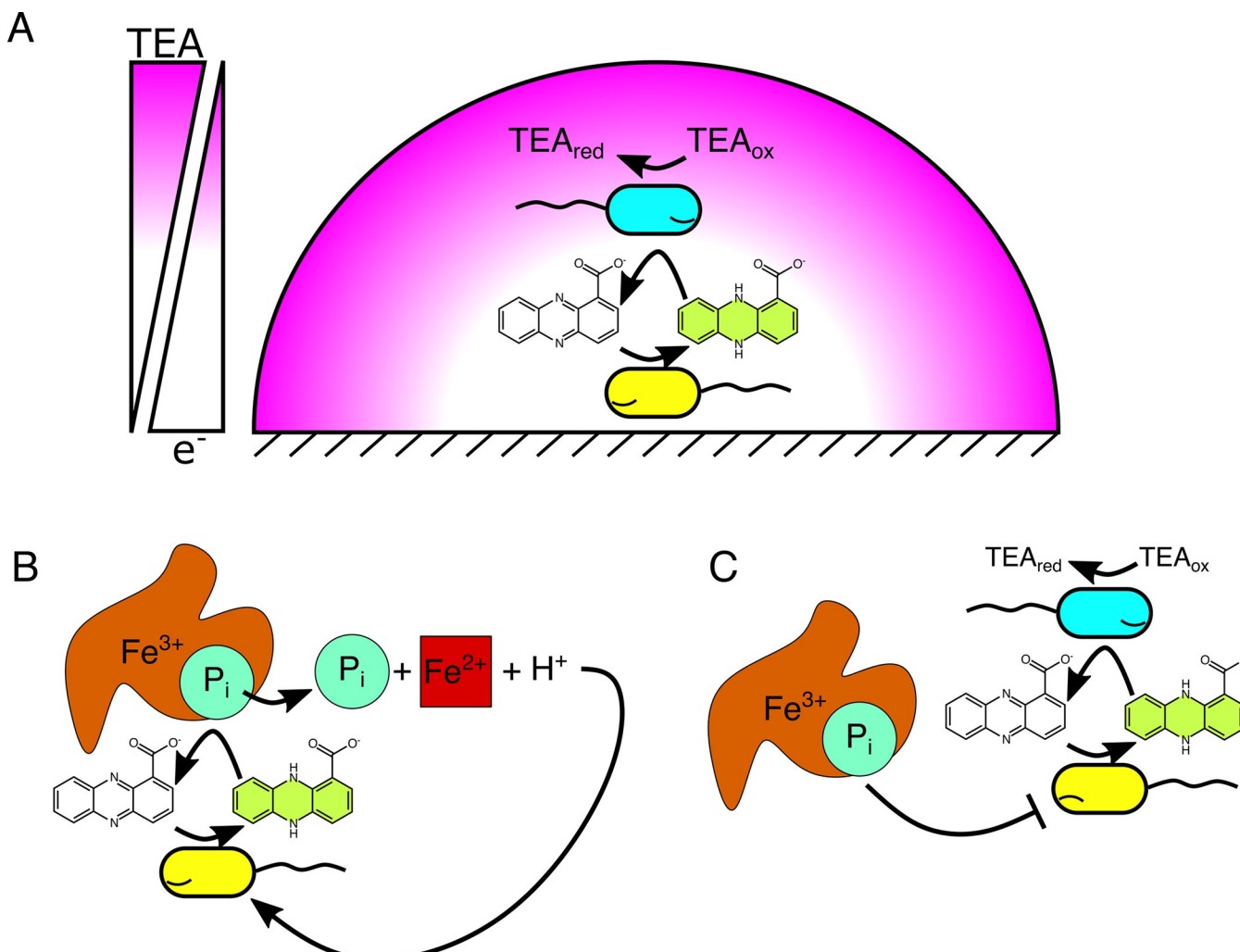

**Fig 8. Model for how phenazine redox cycling may be mutualistic or competitive in microbial communities across redox gradients.** (A) For a bacterium that finds itself starved for its preferred terminal electron acceptor (yellow cell under reducing stress), oxidized PCA can serve as an alternative. For a bacterium that finds itself starved for its preferred electron donor (cyan cell under oxidative stress), reduced PCA can serve as an alternative. Together, the two cells can more efficiently use bioavailable nutrients with the PCA redox cycle as a bridge. This scheme would work best if the phenazine oxidizer has exclusive access to the terminal electron acceptor and that the phenazine reducer has exclusive access to the terminal electron donor. (B-C) For a bacterium that relies on the abiotic oxidation of PCA (yellow cell, B) to release insoluble nutrients from iron minerals such as ferric oxyhydroxides to which phosphate is bound, the presence of a PCA oxidizing species would starve it (cyan cell, C). This situation could occur if the environment contains a TEA that the blue cell can respire, but the yellow cell cannot, and the rate of PCA oxidation by the blue cell exceeds its abiotic oxidation rate by iron minerals. A specific example could pair a *P. aeruginosa* (as the yellow cell) and *C. portucalensis* MBL (as the cyan cell) in the presence of fumarate [23].

bacteria. Identifying these partnerships may enable new model systems for studying bacterial interactions beyond metabolite exchange or antibiotic secretion. Given the redox nature of these interactions, they are worthwhile to explore in bioelectrochemical reactors, where bacterial phenazine oxidation/reduction processes may compete with or supplement electrode activity. A clear direction for future studies is to assess the relative survival or fitness of different *C. portucalensis* MBL genotypes in electrode chambers. For example, a quinone null strain with only one of the three nitrate reductases subjected to a reducing electrode under anoxia would be forced to rely on electrons from PCA to drive nitrate reduction: will this culture survive better than one provided no PCA? No electrode? How might its PCA oxidation activity impact another organism grown in co-culture? The methods and results of this report open

the field to ask detailed questions about the role of electron donors, separate from carbon sources, in bacterial metabolism.

We end by noting that this work was made possible by recent advances in genetic engineering methods, allowing us to take an organism out of soil and develop a full genetic toolkit to test hypotheses in the span of a few years [15,28]. Notably, this work underscores the importance of testing phenotypes across multiple physiological conditions, as some gene activities may be cryptic under a given gene regulatory mode. There are likely many other similar redox metabolisms in the environment that await discovery, and our report provides an example of how they may be pursued.

## Materials and methods

### Reagents

All chemical compounds used were purchased from Sigma-Aldrich. We want to note for future researchers that we encountered batch effects with different lots of TMAO: some lots (particularly the hydrated TMAO stocks) abiotically oxidize PCA while others do not. With TMAO, it may be important to try a couple of batches of the compound.

### Strains and culturing

We employed wildtype and mutant *C. portucalensis* MBL and *P. aeruginosa* UCBPP-PA14 in this study (Table 2). How the mutants were generated is described in the following section and in the supplement (S1 Text). In brief, all strains were cultured in lysogeny broth (LB) prior to PCA oxidation assays. Each pre-growth in 5 mL LB was for 17 hours at 30°C in borosilicate glass culture tubes. In Fig 3, we compare pre-growth in slanted shaking tubes (250 rpm) and standing tubes sealed with parafilm to reduce oxygen permeance. In the bioelectrochemical studies (Figs 7 and S7), wildtype *C. portucalensis* MBL was first pre-grown overnight in shaking slanted tubes and then outgrown to a larger volume before concentrating the cells and inoculating them into the reactors (see the supplement for a detailed protocol, S1 Text). For all other experiments, *C. portucalensis* MBL strains were pre-grown in standing, sealed tubes without supplemented nitrate.

### PCA oxidation assays

The basal PCA oxidation assay medium was composed of 20 mM potassium phosphate buffer (pH 7–7.1); 1 mM sodium sulfate; 10 mM ammonium chloride; and 1× freshwater salt solution (17.1 mM sodium chloride, 1.97 mM magnesium chloride, 0.68 mM calcium chloride, and 6.71 mM potassium chloride). Reduced PCA was prepared in this basal medium at a concentration of 1.2 mM by electrolysis (working electrode poised to -500 mV). Reduced PCA was the only electron donor added to the assay medium. We performed PCA oxidation assays

**Table 2. Strains used in this study.**

| Genotype | Purpose |
|---|---|
| WT, wt, or MBL | The original "wildtype" isolate *C. portucalensis* MBL |
| *menA-tlKO* | Translational knock out of menaquinones and demethylmenaquinones. This is a ubiquinone-only strain |
| *ubiC-tlKO* | Translational knock out of ubiquinones. This is a menaquinone and demethylmenaquinone strain |
| *menAubiC-tlKO* | Translational knock out of all quinones. Abolish the respiratory ETC |
| *napA-tlKO* | Translational knock out of the periplasmic nitrate reductase catalytic subunit |
| *narG-tlKO* | Translational knock out of the dominant respiratory nitrate reductase catalytic subunit |

*(Continued)*

**Table 2.** (Continued)

| Genotype | Purpose |
|---|---|
| *narZ-tlKO* | Translational knock out of the auxiliary respiratory nitrate reductase catalytic subunit |
| *napAnarZ-tlKO* | A NarG-only strain for assessing its individual activity |
| *narGnapA-tlKO* | A NarZ-only strain for assessing its individual activity |
| *narGnarZ-tlKO* | A NapA-only strain for assessing its individual activity |
| *napAnarZnarG-tlKO* | Translational knock out of all the nitrate reductases to assess their total necessity for PCA oxidation |
| *frdA-tlKO* | Translational knock out of the fumarate reductase catalytic subunit |
| *dmsA-tlKO* | Translational knock out of the DMSO reductase catalytic subunit |
| *torA-tlKO* | Translational knock out of the TMAO reductase catalytic subunit |
| *dmsAtorA-tlKO* | Double translational knock out of the DMSO and TMAO reductase catalytic subunits |
| *menAubiCnapAnarZ-tlKO* | Quadruple translational knockout to assess the activity of NarG in the absence of quinones |
| *menAubiC-narGnapA-tlKO* | Quadruple translational knockout to assess the activity of NarZ in the absence of quinones |
| *menAubiCnarGnarZ-tlKO* | Quadruple translational knockout to assess the activity of NapA in the absence of quinones |
| WT/pHelper_Ec1_V1_gentR | The wildtype strain carrying the oligo-mediated recombineering helper plasmid |
| *menA-tlKO*/ pHelper_Ec1_V1_gentR | The *menA* translational knockout strain carrying the oligo-mediated recombineering helper plasmid |
| *menAubiC-tlKO*/ pHelper_Ec1_V1_gentR | The *menAubiC* double translational knockout strain carrying the oligo-mediated recombineering helper plasmid |
| *menAubiCnapA-tlKO*/ pHelper_Ec1_V1_gentR | The *menAubiCnapA* triple translational knockout strain carrying the oligo-mediated recombineering helper plasmid |
| *menAubiCnarG-tlKO*/ pHelper_Ec1_V1_gentR | The *menAubiCnarG* triple translational knockout strain carrying the oligo-mediated recombineering helper plasmid |
| *napA-tlKO*/ pHelper_Ec1_V1_gentR | The *napA* translational knockout strain carrying the oligo-mediated recombineering helper plasmid |
| *napAnarZ-tlKO*/ pHelper_Ec1_V1_gentR | The *napAnarZ* double translational knockout strain carrying the oligo-mediated recombineering helper plasmid |
| *narG-tlKO*/ pHelper_Ec1_V1_gentR | The *narG* translational knockout strain carrying the oligo-mediated recombineering helper plasmid |
| *dmsA-tlKO*/ pHelper_Ec1_V1_gentR | The *dmsA* translational knockout strain carrying the oligo-mediated recombineering helper plasmid |
| *menAubiC-tlKO*/ pFE21-MenA | Inducible expression of MenA in the full quinone knockout background to assess complementation |
| *menAubiC-tlKO*/ pFE21-UbiC | Inducible expression of UbiC in the full quinone knockout background to assess complementation |
| *napAnarZnarG-tlKO*/ pFE21-NapA | Inducible expression of NapA in the full nitrate reductase knockout background to assess complementation |
| *napAnarZnarG-tlKO*/ pFE21-NarZ | Inducible expression of NarZ in the full nitrate reductase knockout background to assess complementation |
| *napAnarZnarG-tlKO*/ pFE21-NarG | Inducible expression of NarG in the full nitrate reductase knockout background to assess complementation |
| *frdA-tlKO*/ pFE21-FrdA | Inducible expression of FrdA in the fumarate reductase knockout background |
| *dmsA-tlKO*/ pFE21-DmsA | Inducible expression of DmsA in the DMSO reductase knockout background |
| MBL/pKD46 | WT strain carrying the helper plasmid for deletion by homologous recombination |
| Δ*napFDAGHBC* | Deletion of the entire Nap operon |
| Δ*narGHJI* | Deletion of the entire NarG operon |
| Δ*narUZYWV* | Deletion of the entire NarZ operon |
| Δ*narZYWV* | Deletion of the entire NarZ operon except for the nitrate-nitrite antiporter NarU |
| PA14 | The wildtype *P. aeruginosa* UCBPP-PA14 strain |
| PA14 Δ*nar* | Δ*narG* deletion in the PA14 strain. Constructed by Steven Wilbert. |
| PA14 Δ*nap* | Δ*napAB* deletion in the PA14 strain. Constructed by Steven Wilbert. |

**Table 3. Oligos for translational knockouts.**

| Name | 5'-3' sequence |
| --- | --- |
| ubiC_MAGE_28–30 | t*t*tggtcatggaatcttccaatTacTaTcagtcgagcagttgggcgtccagcgccgggatcgcatcaaaataacgcagcgcacgcagttg |
| ubiC_MASC_28–30_Rmut1 | gacgcccaactgctcgactgAtAgtA |
| ubiC_MASC_28–30_Rwt1 | gcccaactgctcgactggttgtt |
| ubiC_MASC_28–30_F330 | gccaatctcaataaaatctcgggtcaatgtcg |
| ubiC_28–30_SeqR | gcgatacaatgccttcaggttataaatcgg |
| menA_MAGE_37–39 | g*t*acggcattagcatAgtgATaaggatacttcgatccgctggttgctctgctggcattgattacggcaggactgctgcaaattctgtcca |
| menA_MASC_37–39_Fmut2 | catcgtcggtacggcattagcatAgtgAT |
| menA_MASC_37–39_Fwt2 | cggtacggcattagcatggtggc |
| menA_MASC_37–39_R250 | cagcccggacagacagatcagca |
| menA_37–39_SeqF | atgactgaacaacaacaaattagccgctcac |
| napA_MAGE_92–94 | c*t*gcgttaaacggtcttAtcACtacatgattttcggcaggaagtaacctttgatgcagttaagcccacggttaaccggcgcatccggatc |
| napA_MASC_92–94_Fmut2 | caaaggttacttcctgccgaaaatcatgtaGTgaT |
| napA_MASC_92–94_Fwt2 | ggttacttcctgccgaaaatcatgtacggaa |
| napA_MASC_92–94_R200 | agagccgaacataccgatagactccgg |
| napA_92–94_seqF | cggaacgcagcagggg |
| napA_92–94_seqR | ggtcccagctaatcgggg |
| narZ_MAGE_109–111 | c*a*cgcggcgccagctactcgtggtatctctatagcgctaaccgcTAgTaatGA cctctggtgcgtaaacgcttaatcgaattgtggcgcg |
| narZ_MASC_109–111_Fmut2 | cgattaagcgtttacgcaccagaggTCattAcTA |
| narZ_MASC_109–111_Fwt2 | cgattaagcgtttacgcaccagaggatatttcag |
| narZ_MASC_109–111_R300 | gggcacgggcaagtgatgca |
| narZ_109_111_seqF | tttttgcgggtcgttcataatggatt |
| narZ_109_111_seqR | agatctacgttaaaaacgggctgg |
| natG_MAGE_163–165 | t*g*atggtgtaaacgttagatgcagcaatcagttcattcacttActATcaggaagaacgaacaaatccaccacgaccacgcgcttgtttga |
| narG_MASC_163–165_Fmut2 | cgtggtggatttgttcgttcttcctgATagT |
| narG_MASC_163–165_Fwt2 | gtggatttgttcgttcttcctggcagg |
| narG_MASC_163–165_R200 | agtaccagtcgtagaagctcaggcagg |
| narG_163–165_SeqF | ggtagacgcatgggcatccatcatc |
| frdA_MAGE_121–123 | a*g*cgtcaacgtacgtcgcttcggcggcatgTaaTGATagcgtacctggtttgccgcgggataagaccggcttccacatgctgcatacgctg |
| frdA_MASC_121–123_Fmut | cgcttcggcggcatgTaaTGAT |
| frdA_MASC_121–123_Fwt | cgcttcggcggcatgaaaatcg |
| frdA_MASC_121–123_R343 | cgaattccatatcacgcagcggaacg |
| frdA_121–123_SeqF | tcgcgcaggatcatgacagc |
| dmsA_MAGE_17–19 | c*c*accgccagtccgcctatcgccgtcgttttcaccTaTcAtTAgcgactaacctcagctgccagcatggcatcggggatcttagttttca |
| dmsA_MASC_17–19_Fmut | cctatcgccgtcgttttcaccTaTcAtTA |
| dmsA_MASC_17–19_Fwt | cctatcgccgtcgttttcaccaaacctcg |
| dmsA_MASC_17–19_R244 | gcctacctccataggagtaatactgcc |
| dmsA_17–19_SeqF | tgttgtcggtttcaacatactttatttcaccg |
| torA_MAGE_63–65 | t*a*agcatatcgcagggggtatttgtcatgtttgaacggacgggtttcTTATcactAaccgttgaccactttggcttcaaaagcgccatagt |
| torA_MASC_63–65_Fmut | agccaaagtggtcaacggtTagtgATAA |
| torA_MASC_63–65_Fwt | agccaaagtggtcaacggtgagtggacc |
| torA_MASC_63–65_R200 | gcgcctgatcccaactgaccc |
| torA_63–65_SeqF | gcgatgtgtctgaaaccaacaaaaagtaagg |

Asterisks indicate phosphorothioate bonds between nucleotides. Uppercase nucleotides indicate mutations from the wildtype sequence. MAGE oligos were used to introduce translational knockout mutations. MASC oligos were used to verify mutations by PCR. Seq oligos were used to amplify mutated regions for verification by sequencing.

**Table 4. Oligos for Datsenko-Wanner deletions.**

| Name | 5'-3' sequence |
|---|---|
| Δnap_F | cgagggagggcgtcattatggaaggacaatgtcatgGTGTAGGCTGGAGCTGCTTC |
| Δnap_R | acagtattctgtcatctcgctcgcaaagtaattttaCATATGAATATCCTCCTTAG |
| Δnap_check_F | ataaatagtgcagtttttttatgtgcgttcaacccg |
| Δnap_check_R | acgcgaaatgacagattgctgaacag |
| ΔnarGHJI_F | agagagccgtcaggctcctacaggagaaaaccgatgGTGTAGGCTGGAGCTGCTTC |
| ΔnarGHJI_R | cgaagcagggtttttgtcattctcaaatatacgattaCATATGAATATCCTCCTTAG |
| ΔnarGHJI_check_F | ttcgctctcaatcaagcaatgtcgatttatc |
| ΔnarGHJI_check_R | aaacaggcataaaaaaaccccgccg |
| ΔnarZYWV_F | cgtcggcaatattatcgaagcaggagttatgtcatgGTGTAGGCTGGAGCTGCTTC |
| ΔnarZYWV_R | taagcacaagcgctaccgggcatcagagttcaattaCATATGAATATCCTCCTTAG |
| ΔnarUZYWV_F | tatctattctttcagtcaatattcctaaaactttcGTGTAGGCTGGAGCTGCTTC |
| ΔnarUZYWV_R | aattatcggcgggaacgcactatctgatagcggcgaCATATGAATATCCTCCTTAG |
| ΔnarUZYWV_check_F | ttaagcaacgtgtaattctcccatcacg |
| ΔnarUZYWV_check_R | atcataacgtacaaaaagcgggatcgc |
| KanR_arbF | attcgcagcgcatcgccttctatc |
| λRed_F | tccacattgattatttgcacggcgtcac |
| λRed_R | ccggtgtcatgctgccaccttc |

Lowercase letters in Δ primers indicate homology to the *C. portucalensis* MBL genome. Uppercase letters indicate homology to the pKD4 plasmid.

either by tracking the fluorescence of reduced PCA over time in a plate reader (BioTek Synergy 4 or HTX) housed in a Coy-brand anoxic chamber or by tracking the current generated by a culture when provided PCA and an electrode poised to continuously reduce any PCA that the cells oxidized. Depending on the experiment, 10 mM of the terminal electron acceptor (nitrate, fumarate, DMSO, or TMAO) was added to the assay medium. The current traces were measured using a Gamry potentiostat, and these experiments were conducted in an mBraun-brand anoxic chamber, which scrubs hydrogen from the headspace. A detailed protocol for the plate reader assay is available via protocols.io (dx.doi.org/10.17504/protocols.io. bp2l6xm6dlqe/v1), and an explanation of how our electrode chamber assay differed from a previously published protocol [39] is available in the supplement (S1 Text).

**Table 5. Plasmids employed in the study.**

| Name | Purpose | Reference |
|---|---|---|
| pKD46 | λRed helper, temperature-sensitive origin of replication, confers ampicillin resistance | [19] |
| pKD4 | Kanamycin vector with FRT sites | [19] |
| pCP20 | FLPase vector, temperature-sensitive origin of replication, confers ampicillin resistance | [19] |
| pHelper_Ec1_V1_gentR | Oligo recombineering helper vector, cloned into backbone with SacB, confers gentamicin resistance | [40] |
| pFE21-sfGFP | Tetracycline-inducible promoter driving gene expression, confers kanamycin resistance | [41] |
| pFE21-MenA | pFE21 for inducible MenA expression | This work |
| pFE21-UbiC | pFE21 for inducible UbiC expression | This work |
| pFE21-NapA | pFE21 for inducible NapA expression | This work |
| pFE21-NarZ | pFE21 for inducible NarZ expression | This work |
| pFE21-NarG | pFE21 for inducible NarG expression | This work |
| pFE21-FrdA | pFE21 for inducible NarG expression | This work |
| pFE21-DmsA | pFE21 for inducible DmsA expression | This work |

**Table 6. Oligos for building complementation vectors.**

| Name | 5'-3' sequence |
| --- | --- |
| pZ/FE_backbone_F | caaataaaacgaaaggctcagtcgaaagac |
| pZ/FE_backbone_R | CTAGTAtttctcctctttaatgaattcggtcagtgc |
| pZ/FE_screen_F2 | gacattaacctataaaaataggcgtatcacgagg |
| pZ/FE_screen_R2 | ggcggatttgtcctactcaggagag |
| MenA_pZ/FE_F | accgaattcattaaagaggagaaaTACTAGtttttattggcgctaaatatgactgaacaacaac |
| MenA_pZ/FE_R | gtctttcgactgagcctttcgtttttattgttaaagagaccactgacttaagaaaattccaaagacaaac |
| UbiC_pZ/FE_F | accgaattcattaaagaggagaaaTACTAGatgccacaccctgcgttaac |
| UbiC_pZ/FE_R | gtctttcgactgagcctttcgtttttattgtgtttatcttcctctcagtacaacggccg |
| NapA_pZ/FE_F | accgaattcattaaagaggagaaaTACTAGtgagcaaggtgaggaaacaccatgaaac |
| NapA_pZ/FE_R2 | gtctttcgactgagcctttcgtttttattgcagaaaagcggcggcgg |
| NarZ_pZ/FE_F | accgaattcattaaagaggagaaaTACTAGatgagtaaactgttagaccgctttcgc |
| NarZ_pZ/FE_R | gtctttcgactgagcctttcgtttttattgtcattttttcgcctcctgtacctgatc |
| NarG_pZ/FE_F | accgaattcattaaagaggagaaaTACTAGaggagaaaaccgatgagtaaattcctggac |
| NarG_pZ/FE_R | gtctttcgactgagcctttcgtttttattgtcattttacgctctcctgtacctggtc |
| FrdA_pZ/FE_F | accgaattcattaaagaggagaaaTACTAGatgaagcatctgatttcaggcaggg |
| FrdA_pZ/FE_R | gtctttcgactgagcctttcgtttttattgttatagcgcaccacctcaactttcagg |
| dmsA_pZ/FE_F | accgaattcattaaagaggagaaaTACTAGatgaaaactaagatccccgatgccatgc |
| dmsA_pZ/FE_R | gtctttcgactgagcctttcgtttttattgttacacctttttcaacctgaacaaggttcg |

Uppercase nucleotides indicate a modified spacer sequence between the ribosomal binding site and start codon. The first 30 nucleotides at the 5' end of each oligo starting with a gene name are homologous to the pFE21 plasmid to place the gene between the tetracycline-inducible promoter and the terminator in the correct orientation.

### Genetic engineering of C. portucalensis MBL

Detailed protocols for genetically engineering *C. portucalensis* MBL are available in the supplement (S1 Text). In brief, we employed both λRed-mediated homologous recombination to delete endogenous operons [19] and oligo-mediated recombineering to cause translational knockouts of genes of interest [18]. In the λRed approach, the operons were replaced by a kanamycin resistance cassette by homologous recombination using the λRed recombinase expressed on a transient plasmid [19]. The oligonucleotides and primers we used to accomplish this are listed in Tables 3 and 4. For generating the deletion strains, we used the helper plasmids pKD46 (for the λRed machinery), pKD4 (for the kanamycin resistance cassette flanked by FRT sites), and pCP20 (for the FLPase).

In the oligo-mediated recombineering approach, we replaced three subsequent codons with the three different stop codons (TAA, TGA, and TAG, not necessarily in that order) in the first half of the coding sequence of the catalytic subunit of the given gene(s). We used the helper plasmid pHelper_Ec1_V1_gentR, which was derived from pORTMAGE-Ec1 [18,40] by adding a SacB gene for sucrose counterselection to cure the plasmid. These helper plasmids are listed in Table 5. We transformed *C. portucalensis* MBL by electroporation: 200 Ω, 25 μF, 2.5 kV in 2 mm gap cuvettes. Our protocols for preparing electrocompetent *C. portucalensis* MBL, the Datsenko-Wanner knockouts, and the oligo-mediated recombineering are available via protocols.io:

1. Electrocompetent cells: dx.doi.org/10.17504/protocols.io.kqdg3x7r7g25/v1

2. Datsenko-Wanner knockouts: dx.doi.org/10.17504/protocols.io.ewov1q5e7gr2/v1

3. Oligo-mediated recombineering: dx.doi.org/10.17504/protocols.io.eq2lyj1rwlx9/v1

## Cloning and induction of complementation vectors

We cloned the complementation vectors using the plasmid pFE21 as a backbone for inducible expression of wildtype *C. portucalensis* MBL genes [41]. We used Gibson assembly to construct the new vectors from the pFE21 backbone and gene amplicons, and the primers we used are listed in Table 6. We induced the vector by incubating the cultures with 50 nM anhydrotetracycline during the pre-growth phase of the assay.

## Supporting information

**S1 Fig. KEGG reference pathway for quinone biosynthesis (map00130).** The relevant genes are indicated by red boxes and the relevant quinones by red ellipses. Note: *MenG* is a homolog to *UbiE* from photosynthetic organisms and is not present in γ-Proteobacteria like *C. portucalensis* MBL; the alternative pathway for menaquinone biosynthesis via futalosine (the *Mqn* genes) is also absent in *C. portucalensis* MBL [42–44]. Loss of *UbiC* results in the loss of ubiquinones. Loss of *MenA* results in the loss of menaquinones and demethylmenaquinones. Loss of *UbiE* results in the loss of ubiquinones and demethylmenaquinones. Note: *Pseudomonas aeruginosa* only has ubiquinones in its ETC under both aerobic and anaerobic growth conditions [31]. Pathway diagram used with permission from Kanehisa Laboratories (permission received November 6, 2023)[45].
(TIF)

**S2 Fig. All pairwise comparisons of PCA oxidation dynamics of nitrate reductase knockouts, including shaking (oxic) versus standing (hypoxic) overnight pregrowth.** (A) Pairwise comparisons of the maximum PCA oxidation rate by all nitrate reductase genotypes after oxic pregrowth. This corresponds to Fig 3D. (B) Pairwise comparisons of the maximum PCA oxidation rate by all nitrate reductase genotypes after hypoxic pregrowth, corresponding to Fig 3E. (C) All pairwise comparisons for the time to oxidize half of the provided PCA, corresponding to Fig 3F. (D) Finally, all the pairwise comparisons for the half-max oxidation time for Fig 3G.
(TIF)

**S3 Fig. Quality of LOWESS fit depends on the parametrization of its scanning window.**
(A-B) A demonstration of the analysis pipeline. (A) We used a locally weighted scatterplot smoothing (LOWESS) algorithm to fit a curve to our empirical data, here showing an example of one of the biological replicates for wildtype *C. portucalensis* MBL oxidizing PCA with nitrate after hypoxic overnight pregrowth. The data are in semitransparent circles and the LOWESS fit is the red line. The fitting was parametrized to use 5% of the data in its sliding window. This fit allowed us to determine the time it took the cultures to oxidize half of the provided PCA ($T_{half\ max}$, or any arbitrary threshold) and the derivative, or rate, of PCA oxidation. (B) This derivative allowed us to estimate the maximum rate of PCA oxidation and the time at which it occurred. The output of the LOWESS algorithm depends on one key parameter: the fraction of data that it considers for each smoothing window. (C) Examples of different fits to the same data as in Fig 4A, scanning the fraction parameter from 0 to 1. The value that was used for the analysis, 0.05, is in red. (D) How the estimated maximum PCA oxidation rate depends on the scanning window. The red square indicates the value at 0.05. (E) How the time to half of PCA being oxidized depends on the scanning window. The red square indicates the value at 0.05. (F) A 2D representation of both the maximum oxidation rate and the time at which it occurs. The red square indicates the value at 0.05. 0.05 was chosen as the window for all the analyses in this report as it appeared to give stable outputs while using a minimal window for fitting.
(TIF)

**S4 Fig. Complementation of nitrate reductase knockouts after overnight standing pre-growth.** (A) The maximum PCA oxidation rate for the triple knockout strain and overexpressed individual nitrate reductases in that genetic background. Squares represent the means of technical triplicates and circles represent independent biological replicates. The negative value in the triple knockout background indicates that the cells were further reducing the provided stock of PCA, rather than oxidizing it. (B) Pairwise statistical tests against the null hypothesis that there is no difference between the mean maximum oxidation rates of two given genotypes. Given six comparisons, the Bonferroni-corrected p-value threshold for significance is $p < 0.00833$.
(TIF)

**S5 Fig. Overexpression of MenA and UbiC does not complement the PCA oxidation phenotype of the menAubiC double mutant.** (A) Maximum PCA oxidation rates in complemented quinone knockout backgrounds. (B) Pairwise comparisons of the mean maximum PCA oxidation rates in (C). Given six comparisons, the Bonferroni-corrected threshold for significance is $p < 0.00833$.
(TIF)

**S6 Fig. All pairwise comparisons for the strains in Fig 6.** (A) Statistical significance matrix for the maximum PCA oxidation rate in the presence of fumarate. (B) Statistical significance matrix for the maximum PCA oxidation rate in the presence of DMSO. (C) Statistical significance matrix for the maximum PCA oxidation rate in the presence of TMAO.
(TIF)

**S7 Fig. PCA oxidation depends on nitrate availability in a bioelectrochemical reactor.** Each chart presents a time course of current (μA on a linear scale) during incubation of *C. portucalensis* MBL in a bioelectrochemical reactor with a working electrode that continuously reduces PCA. Current indicates that the culture is oxidizing PCA. Vertical black bars represent the timing of nitrate spiking, when appropriate. Each chart is titled according to the initial concentration of nitrate in the medium and the spiking schedule.
(TIF)

**S1 Text. Supplementary results.**
(DOCX)

## Acknowledgments

We thank Steven Wilbert, John Ciemniecki, Chelsey VanDrisse, Georgia Squyres, Avi Flamholz, and Julian Wagner for helpful technical feedback and general support throughout this work. We thank Maxim Tsypin for pointing us to Efron and Tibshirani's implementation of the bootstrapped hypothesis test.

## Author Contributions

**Conceptualization:** Lev M. Z. Tsypin, Dianne K. Newman.

**Data curation:** Lev M. Z. Tsypin.

**Formal analysis:** Lev M. Z. Tsypin.

**Funding acquisition:** Lev M. Z. Tsypin, Dianne K. Newman.

**Investigation:** Lev M. Z. Tsypin, Allen W. Chen.

**Methodology:** Lev M. Z. Tsypin, Scott H. Saunders, Dianne K. Newman.

**Project administration:** Lev M. Z. Tsypin, Dianne K. Newman.

**Resources:** Dianne K. Newman.

**Software:** Lev M. Z. Tsypin.

**Supervision:** Lev M. Z. Tsypin, Scott H. Saunders, Dianne K. Newman.

**Validation:** Lev M. Z. Tsypin.

**Visualization:** Lev M. Z. Tsypin.

**Writing – original draft:** Lev M. Z. Tsypin.

**Writing – review & editing:** Lev M. Z. Tsypin, Scott H. Saunders, Allen W. Chen, Dianne K. Newman.

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
