## [Decision Letter · Decision Letter 0]

22 Jan 2024

Dear Dr Newman,

Thank you very much for submitting your Research Article entitled 'Genetically dissecting the electron transport chain of a soil bacterium reveals a generalizable mechanism for biological phenazine-1-carboxylic acid oxidation' to PLOS Genetics.

The manuscript was fully evaluated at the editorial level and by independent peer reviewers. The reviewers appreciated the attention to an important problem, but raised some substantial concerns about the current manuscript. Based on the reviews, we will not be able to accept this version of the manuscript, but we would be willing to review a much-revised version. We cannot, of course, promise publication at that time.

If you decide to revise the manuscript for further consideration at PLOS Genetics, please aim to resubmit within the next 60 days, unless it will take extra time to address the concerns of the reviewers, in which case we would appreciate an expected resubmission date by email to plosgenetics@plos.org.

We are sorry that we cannot be more positive about your manuscript at this stage. Please do not hesitate to contact us if you have any concerns or questions.

Yours sincerely,

Jue D. Wang

Academic Editor

PLOS Genetics

Lotte Søgaard-Andersen

Section Editor

PLOS Genetics

Reviewer's Responses to Questions

**Comments to the Authors:**

Reviewer #1: In this paper, Tyspin et al., report studies to provide genetic evidence for biological oxidation of phenazine-1-carboxylaic acid (PCA) a compound with previously reported ability to support extracellular electron transfer in bacteria. It is well known that reduced PCA can participate in extracellular electron transfer, but the relevant abiotic and biological pathway(s) for its oxidation are largely unknown. In this work the authors perform experiments to analyze the relative contribution of abiotic PCA oxidation, the ability of PCA to reduce bacterial quinones to their respective quinols, or the ability of reduced PCA to donate electrons to one or more alternative respiratory chains that are often only present or more abundant under anaerobic conditions. This is an important and unresolved question in the field that often does not have well-controlled studies or makes conclusions without the needed supporting data.

For these above reasons, it is my opinion that the findings of this work should be of interest to many readers and could set the stage for many important subsequent studies. However, the current version of the manuscript lacks important information that will allow many readers to appreciate the results and allow others to repeat many of the experiments. Thus, it is my opinion that some major rewriting, revisions, deletions and additions to the paper are needed.

1. The presentation of the work makes it difficult for the reader to follow and obtain the major conclusions of the work. For example, the focus on Category 1, 2 and 3 routes for PCA oxidation may be obvious to the authors but this reader found themselves constantly referring to the text to remember what each of these categories were. The text should simply state abiotic oxidation, PCA reduction of quinone, or transfer to an alternative terminal oxidase.

2. The Results section reads more like an introduction to the work or a Discussion of the data than a presentation of the experimental data. On one hand it takes almost 3 pages before an experiment is described and results presented. In addition, there is little if any reference to primary data in this section to guide the reader through the text and help them understand how the experimental data leads to conclusions.

3. It is a concern that I can not find anything in the paper (M&M or supplementary) that reports the amount of alternative electron acceptor present in the media. Without this information, others will not be able to reproduce the experimental design or data in this paper.

4. The data presentation also makes it hard for others to reproduce experiments in the future. There is little primary data (growth curves, primary data for rates of PCA oxidation, quinone reduction etc) presented for the wild type or mutant strains of both organisms. Instead relative data are presented with not information on actual data. Also in the studies with mutants, the reader does not know if the difference in the rate of PCA oxidation (by any route) reflects a difference in the growth of the cultures when a quinone biosynthetic or alternative electron transfer pathways if blocked.

5. The Discussion and interpretation of the data is often based on the “assumption” that the control of the alternative election acceptor enzymes in these organisms will be the same or at least very similar to that in E. coli. With the advent of genomics, it is clear that the oxygen regulation of alternative electron transfer pathways is often not the same. Thus, I would caution the authors about making conclusions in the Discussion without additional experiments, in subsequent papers.

6. I may have mis-understood the logic (for reasons mentioned above), but I could make a case that PCA reduction of quinone is as important, or maybe even more important, that the use of PCA as an electron donor to one or more of the alternative oxidases based on the analysis of mutants in both species.

7. In the case of PCA reduction of quinones, can a single electron transfer from reduced PCA can generate a semi-quinone. Semi-quinone formation could generate reactive intermediates and a stress response that could impact how one interprets the data.

8. Along these same lines, it is possible that electron transfer to nitrate can generate RNS that case a stress response and complicate the interpretation of the data?

Reviewer #2: In this manuscript, Tsypin et al. dissect the molecular mechanisms underlying biological phenazine oxidation in Citrobacter portucalensis MBL and Pseudomonas aeruginosa. They develop new genetic tools for C. portucalensis MBL over the course of the study to also establish this strain as a new model to study extracellular electron transfer processes in microorganisms.

This is a careful, detailed and foundational study that establishes a genetic framework for biological phenazine oxidation and its physiological impact on microbes. In terms of presentation, the experimental design and the underlying rationale were conveyed beautifully and the data visualization is on point. I also want to commend the rigorous and thoughtful statistical analyses conducted throughout this study. My only criticism is that the paper is far too long and meandering (especially in the results section) and needs to be condensed substantially to make it punchier and easier to read. A few suggestions to condense the paper are listed below:

• Move the theoretical framework to the supplement (lines 131-208). While it is really important, and should be a part of this paper, the details are somewhat tangential. The authors can mention which reactions have a positive or negative ∆G in the main text and refer to the supplement for more details.

• Refrain from describing all pairwise comparisons for mutant strains. While I appreciate the comprehensive genetic analyses that is conducted in this study, I was starting to get a little lost in the weeds when all the mutants were compared to each other under a number of different conditions. It would be preferable to only describe the phenotypes of mutants that are important for the story and let the readers infer the rest from the figures. Less is more!

• Move the description of the statistical analyses to the supplement. Again, a really important part of the paper but the details are tangential to the main storyline and, as a result, distracting. I would also recommend moving this section to the supplement and referring to it in the main text or the figure legends as appropriate.

• Refrain from repeating results in the discussion. The discussion can be condensed to describe an overarching framework for PCA oxidation and its relevance in the context of the microbe and the environment as well as the limitations of the study. Details re: specific fellow up experiments are nice but not as useful.

Additional, more specific and minor comments are listed below:

• Lines 246-253 is more relevant for the methods section. Since one type of genetic perturbation system is used after an initial trial, it might only be worth bringing it up in the methods, just to streamline things a bit.

• Figure 3 and Figure 4 can be combined. I think Figure 4 panels A, B, E, and F can all be removed, and the rates in panels B and C can be combined with Figure 3. Additionally, I think the “time to half oxidation” plots are a very interesting piece of data that can be moved out of the Sup into main text. I would include Sup Fig 4.2 panels A and B in the new version of Figure 3 that has the time course experiments and the rate plots. From there, you can explain that only rates will be used in the next few figures. Also, figure 4 legend contains information that should be only in the main text. Keep analysis of the results in the text.

• Figure 5 and the associated quinone analysis is very informative. However, I think the paper is lacking justification as to why a UbiE mutant was not made. This would be the most important mutant to make to separate out and rank which quinones contribute the most to the oxidation of PCA. This should be discussed more in the discussion.

• Line 421 – “exacerbated” should be “marginally rescued” maybe?

• Lines 446-447, I would emphasize the regulatory mechanisms of each nitrate reductase in P. aeruginosa are opposite in C. portucalensis. This will help drive home that the phenotypes make sense later when the regulatory pathways are discussed in more detail.

• Figure 6, can the abiotic control line also be included? Was one done in this experiment?

• Figure 7, remove panels C, F, I. Move stats to supp. Keep mutants in the same order across panels for ease of reading

• Line 681 – “Figure 12” is a typo

In terms of the scientific content, I have a few conceptual questions

1) The authors repeatedly mention that PCA oxidation might be beneficial under anaerobic conditions where electron donors are limited. I am confused by this statement as I would expect that fermentation, which is more common under anaerobic conditions, would lead to the excretion of reduced organic compounds that can readily serve as electron donors. If the authors can provide any evidence that electron donors are indeed limiting under anaerobic conditions or in anoxic environments that would be useful to bolster some of the central claims re: the relevance of PCA oxidation in this study.

2) I remain confused about the mechanistic details of PCA oxidation coupled to nitrate reduction based on the genetic evidence presented in this study. The authors clearly demonstrate that a triple KO of C. portucalensis lacking all three nitrate reductases is indistinguishable from the abiotic control (Figure 4C and 4D). The authors also show that a menAubiC KO can barely oxidize PCA (Figure 5A). These data, to me, are consistent with two alternate hypotheses:

• The nitrate reductase complex is responsible for oxidizing PCA and requires an input of electrons from the quinone pool to do so – not sure how and why?; in the absence of the nitrate reductase complex, the quinone pool can oxidize PCA minimally, likely because it is difficult for PCA (hydrophilic) to access quinones in the membrane

• The quinone pool oxidizes PCA and requires a functional nitrate reductase complex to regenerate itself; in the absence of quinones maybe the nitrate reductase complex has some residual capacity to oxidize PCA

I am not quite sure if it is possible to distinguish between these two hypotheses purely based on genetic data. My impression from reading the discussion is that the authors favor the latter hypothesis over the former but I am not quite sure what the underlying reason might be. It would be nice if, in the discussion, the authors state various plausible hypothesis that are consistent with their genetic data for PCA oxidation.

3) None of the experiments in this study address whether PCA oxidation coupled to nitrate/fumarate/DMSO/TMAO reduction facilitate growth or energy conservation in the organism. While the bioreactor experiment in figure 8 show that PCA oxidation enhances survival, the underlying basis is unclear. It would be useful if the authors could perhaps discuss how cells produce ATP/NADH during PCA oxidation coupled to nitrate reduction and whether C. portucalensis can couple this process with CO2 fixation or heterotrophic growth.

4) For the bioreactor experiments, I feel like there is a control missing from this analysis with a reacto

---

## [Decision Letter · Decision Letter 1]

25 Mar 2024

Dear Dr Newman,

We are pleased to inform you that your manuscript entitled "Genetically dissecting the electron transport chain of a soil bacterium reveals a generalizable mechanism for biological phenazine-1-carboxylic acid oxidation" has been editorially accepted for publication in PLOS Genetics. Congratulations!

Yours sincerely,

Jue D. Wang

Academic Editor

PLOS Genetics

Lotte Søgaard-Andersen

Section Editor

PLOS Genetics

Comments from the reviewers (if applicable):

Reviewer's Responses to Questions

**Comments to the Authors:**

Reviewer #1: In my view the authors have either made all the changes requested in my previous review or provided needed responses in their rebuttal. Based on this my recommendation is to accept the paper as revised

**Have all data underlying the figures and results presented in the manuscript been provided?**

Reviewer #1: Yes

PLOS authors have the option to publish the peer review history of their article (what does this mean?). If published, this will include your full peer review and any attached files.

Reviewer #1: No

**Data Deposition**

http://datadryad.org/submit?journalID=pgenetics&manu=PGENETICS-D-23-01276R1

**Press Queries**

---

## [Editor Report · Acceptance letter]

30 Apr 2024

PGENETICS-D-23-01276R1 

Genetically dissecting the electron transport chain of a soil bacterium reveals a generalizable mechanism for biological phenazine-1-carboxylic acid oxidation 

Dear Dr Newman, 

We are pleased to inform you that your manuscript entitled "Genetically dissecting the electron transport chain of a soil bacterium reveals a generalizable mechanism for biological phenazine-1-carboxylic acid oxidation" has been formally accepted for publication in PLOS Genetics! Your manuscript is now with our production department and you will be notified of the publication date in due course.

With kind regards,

Judit Kozma

PLOS Genetics

On behalf of:
